# Immune mediation of HMG-like DSP1 via Toll-Spätzle pathway and its specific inhibition by salicylic acid analogs

**Md. Mahi Imam Mollah**[ID], **Shabbir Ahmed**[ID], **Yonggyun Kim**[ID]*

Department of Plant Medicals, College of Life Sciences, Andong National University, Andong, Korea

* hosanna@anu.ac.kr

**Data Availability Statement:** All relevant data are within the manuscript and its Supporting Information files.

## Abstract

*Xenorhabdus hominickii*, an entomopathogenic bacterium, inhibits eicosanoid biosynthesis of target insects to suppress their immune responses by inhibiting phospholipase $A_2$ ($PLA_2$) through binding to a damage-associated molecular pattern (DAMP) molecule called dorsal switch protein 1 (DSP1) from *Spodoptera exigua*, a lepidopteran insect. However, the signalling pathway between DSP1 and $PLA_2$ remains unknown. The objective of this study was to determine whether DSP1 could activate Toll immune signalling pathway to activate $PLA_2$ activation and whether *X. hominickii* metabolites could inhibit DSP1 to shutdown eicosanoid biosynthesis. Toll-Spätzle (Spz) signalling pathway includes two Spz (*SeSpz1* and *SeSpz2*) and 10 Toll receptors (*SeToll1-10*) in *S. exigua*. Loss-of-function approach using RNA interference showed that SeSpz1 and SeToll9 played crucial roles in connecting DSP1 mediation to activate $PLA_2$. Furthermore, a deletion mutant against *SeToll9* using CRISPR/Cas9 abolished DSP1 mediation and induced significant immunosuppression. Organic extracts of *X. hominickii* culture broth could bind to DSP1 at a low micromolar range. Subsequent sequential fractionations along with binding assays led to the identification of seven potent compounds including 3-ethoxy-4-methoxyphenol (EMP). EMP could bind to DSP1 and prevent its translocation to plasma in response to bacterial challenge and suppress the up-regulation of $PLA_2$ activity. These results suggest that *X. hominickii* inhibits DSP1 and prevents its DAMP role in activating Toll immune signalling pathway including $PLA_2$ activation, leading to significant immunosuppression of target insects.

## Author summary

Immune responses of insects are highly effective in defending various entomopathogens. *Xenorhabdus hominickii* is an entomopathogenic bacterium that uses a pathogenic strategy of suppressing host insect immunity by inhibiting phospholipase $A_2$ ($PLA_2$) which catalyzes the committed step for eicosanoid biosynthesis. Eicosanoids mediate both cellular and humoral immune responses in insects. This study discovers an upstream signalling pathway to activate $PLA_2$ in response to bacterial challenge. Se-DSP1 is an insect homolog of vertebrate HMGB1 that acts as a damage-associated molecular pattern. Upon bacterial

**Funding:** This work was supported by a grant (No. 2017R1A2133009815) of the National Research Foundation (NRF) funded by the Ministry of Science, ICT and Future Planning, Republic of Korea to YK. The funders had no role in study design, data collection and analysis, decision to publish, or preparation of the manuscript.

**Competing interests:** The authors have declared that no competing interests exist.

infection, Se-DSP1 is released to the circulatory system to activate Spätzle, an insect cytokine that can bind to Toll receptor. Toll immune signalling pathway can activate antimicrobial peptide gene expression and $PLA_2$. A deletion mutant against a Toll gene abolished immune responses mediated by Se-DSP1. Indeed, *X. hominickii* can produce and secrete secondary metabolites including salicylic acid analogs that can strongly bind to Se-DSP1. These bacterial metabolites prevented the release of Se-DSP1, which impaired the activation of $PLA_2$ and resulted in a significant immunosuppression of target insects against bacterial infection.

## Introduction

*Xenorhabdus hominickii* is an entomopathogenic bacterium that is mutualistic to *Steinernema monticolum*, a nematode [1]. In general, infective juveniles (IJs) of host nematodes carry pathogenic bacteria in specific receptacles of intestine [2,3]. After entering target insect's hemocoel through anus, spiracle, and mouth, IJs will release symbiotic bacteria to induce insect immunosuppression [4].

Upon pathogen infection, insects can recognize the nonself with their specific molecular patterns such as lipopolysaccharide, peptidoglycan, or β-1,3-glucan using pattern recognition receptors [5]. The nonself signal is then propagated to nearby immune effectors via various immune mediators such as cytokine, biogenic monoamine, nitric oxide (NO), and eicosanoids [6]. Subsequently, acute cellular immune responses are triggered by hemocytes. Later, humoral immune responses can remove pathogens with melanin formation through the catalysis activity of plasma phenoloxidase or the activity of antimicrobial peptides (AMPs) [4].

For successful pathogenicity, *X. hominickii* needs to produce and release secondary metabolites such as phenylethylamides to inhibit phospholipase $A_2$ ($PLA_2$) and shutdown eicosanoid biosynthesis [7,8]. $PLA_2$ catalyzes hydrolysis of phospholipids at *sn-2* ester bond to release arachidonic acid. This is a committed step for eicosanoid biosynthesis in insects as well as in other organisms [9]. Immune challenge can activate $PLA_2$ to produce eicosanoids in insects [10]. Thus, inhibiting te activity of $PLA_2$ by secondary metabolites of *X. hominickii* will lead to immunosuppression, which is favorable for these bacteria to express potent pathogenicity.

Dorsal switch protein 1 (DSP1) plays a crucial role in activating $PLA_2$ in *Spodoptera exigua*, a lepidopteran insect, in response to immune challenge [11]. DSP1 is a homolog of vertebrate high mobility group protein 1 (HMGB1) that is ubiquitously expressed and localized in the nucleus to bind to DNA for transcriptional regulation and chromatin organization [12]. It is released passively from dead cells or actively from activated immune cells, enterocytes, hepatocytes, and other cells [13]. Released HMGB1 can act as a damage-associated molecular pattern (DAMP) molecule and activate innate immune responses [14]. In insects, DSP1 in the nucleus can act as a corepressor of Dorsal protein in *Drosophila melanogaster* [15]. In *Aedes aegypti* (a mosquito), DSP1 can facilitate chromatin remodelling for Toll-associated transcriptional factor to bind to promoter in response to immune challenge [16]. In *S. exigua*, DSP1 is released to plasma upon bacterial challenge. It can activate $PLA_2$ to mediate various immune responses [11]. Toll immune signalling is likely to serve as a functional link between DSP1 and $PLA_2$ activation because Pelle kinase, an adaptor molecule of Toll receptor, is required for $PLA_2$ activation in *S. exigua* upon bacterial challenge [17]. Furthermore, HMGB1 in mammals is known to mediate immune signals via Toll-like receptors [18].

In insects, Toll receptor is activated by binding to its ligand, Spätzle, which is activated by sequential activation of serine proteases after pathogen recognition [19]. This led us to pose a

hypothesis that DSP1 acts as an initial immune inducer by activating the Toll-Spätzle signal pathway to activate PLA$_2$. This hypothesis suggested that *X. hominickii* could suppress DSP1 to inhibit PLA$_2$ activation using its secondary metabolites. For screening secondary metabolites, we used a purified recombinant DSP1 protein to test its binding to a specific metabolite. Chemical identification of DSP1-binding compound supported the hypothesis of DSP1 mediation of PLA$_2$ activity via the Toll-Spätzle signal pathway.

## Results

### Se-DSP1 is released to plasma in exosome cargo

An immunofluorescence assay showed that Se-DSP1 was localized in the nuclei of fat bodies collected from naïve larvae (Fig 1A). When larvae were challenged with a Gram-postive bacterium, *Enterococcus mundtii*, an entomopathogen, Se-DSP1 was detected in the nuclei at less

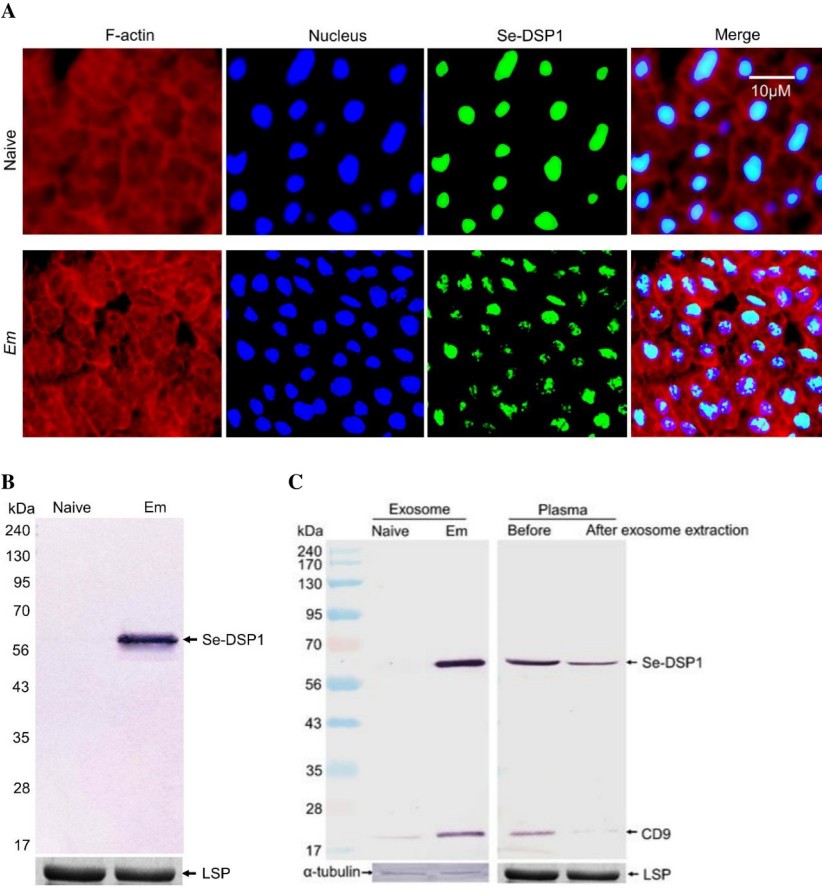

**Fig 1. Secretion of Se-DSP1 from nuclei to plasma upon infection to Gram-positive bacterium, *E. mundtii* ('Em', $4 \times 10^5$ cells/larva), in *S. exigua*.** (A) An immunofluorescence assay of Se-DSP1 in fat body at 6 h after bacterial injection. F-actin and nucleus were stained with phalloidin and DAPI, respectively. Se-DSP1 was detected with its polyclonal antibody. (B) Western blotting analysis of Se-DSP1 in the plasma of naïve or Em-challenged larvae. Plasma samples were collected at 6 h after bacterial challenge. Each lane was loaded with 5 µL of plasma. (C) Exosome analysis for secreted Se-DSP1 in the plasma using western blotting against CD9 (an exosome-specific protein) and Se-DSP1. (Left panel) Exosomes isolated from plasma of naïve or Em-challenged larvae. (Right panel) Plasma samples of Em-challenged larvae before and after exosome extraction. Each lane was loaded with 20 µg proteins. Coomassie-staining bands against a larval storage protein ('LSP') indicated the same amount of protein loading for plasma samples. A cytoskeletal protein, α-tubulin, was detected by western blotting to indicate the same amount of protein loading in exosome analysis.

intensity ($t$ = 5.67; df = 4; $P$ = 0.0048) than in those of naïve larvae, in which relative fluorescence intensity per cell was 92.8 ± 6.94 in naïve larvae and 56.7 ± 8.58 in immune-challenged larvae. An immunoblotting analysis showed that some Se-DSP1 were detected in the plasma of immune-challenged larvae, but not in the plasma of naïve larvae (Fig 1B). When exosomes were extracted from the plasma of immune-challenged larvae, Se-DSP1 was detected in these exosomes which reacted with CD9, an exosome-specific protein (Fig 1C). Plasma after removing exosomes still reacted with an antibody specific to Se-DSP1, indicating the presence of unbound free form of Se-DSP1 in the plasma.

## Se-DSP1 alone mediates immune responses induced by a bacterial challenge

The release of Se-DSP1 to plasma in response to challenge by Gram-positive bacteria suggested that Se-DSP1 alone without bacterial infection might mediate immune responses usually induced by such bacterial challenge. Before testing this hypothesis, we showed that the bacterial pathogen induced immune responses of *S. exigua* in the present study (Fig 2). Phenoloxidase (PO) activity was significantly ($p < 0.05$) induced by bacterial challenge (Fig 2A). PO activity was also significantly induced after injecting a recombinant Se-DSP1 (rSe-DSP1), but not by injecting a heat-inactivated rSe-DSP1. However, RNAi against *Se-DSP1* expression suppressed the induction of PO activity after bacterial challenge, suggesting a role of Se-DSP1 in mediating PO activation. Eicosanoids are known to mediate immune responses [4]. To monitor eicosanoid biosynthesis, activity of PLA$_2$, the enzyme for the synthesis of eicosanoid, was measured after bacterial or rSe-DSP1 injection (Fig 2B). Activities of both sPLA$_2$ and cPLA$_2$ were significantly ($p < 0.05$) induced by either bacterial or rSe-DSP1 injection. However, heat-inactivated rSe-DSP1 did not induce PLA$_2$ activities. Expression levels of AMP genes were also significantly upregulated after bacterial or rSe-DSP1 injection (Fig 2C). When 11 AMP genes were assessed, bacterial challenge significantly ($p < 0.05$) induced 10 AMP genes except *attacin-2*. Especially, expression levels of *apolipophorin-III*, *cecropin*, *defensin*, *gallerimycin*, *gloverin*, and *lysozyme* were increased more than five-fold increases after bacterial challenge. rSe-DSP1 injection induced all AMP genes assessed, with expression levels of gallerimycin, gloverin, and lysozyme being up-regulated by more than five-fold. Some of these AMP genes were induced even by heat-inactivated rSe-DSP1. However, their up-regulated levels were much less than five-fold and regarded as a stress response to a physical damage by injection. There was a high correlation ($r$ = 0.659; $p < 0.027$) between the up-regulation of AMP genes between after bacterial challenge and that after rSe-DSP1 injection. Expression levels of three AMP genes (*Gal*, *Glv*, and *Lyz*) were highly induced by either bacterial challenge or rSe-DSP1 injection.

## Toll-Spz signalling pathway is functionally linked with DSP1

Toll immune signalling pathway is mainly triggered by Gram-positive bacteria [20]. Toll immune signalling is operating in *S. exigua* [21]. Thus, we analyzed Toll signalling in response to Se-DSP1. Ten Toll genes (*SeToll1 ~ SeToll10*) were obtained from *S. exigua* genome (GenBank accession number: WNNL 01000015.1). Of them, at least eight *SeToll* genes were localized on the same chromosome (Fig 3A). All SeTolls were predicted to be transmembrane proteins. Most of them possessed Toll/interleukin-1 receptor homology domain except for SeToll7 (Fig 3B). Phylogeny tree suggested that *SeToll9* and *SeToll7* were closely related to each other, different from other *SeToll* genes. All predicted SeToll genes were expressed in *S. exigua* (S1 Fig). Among these ten Toll genes, *SeToll3* was predominantly expressed in the fat body (Fig 3C).

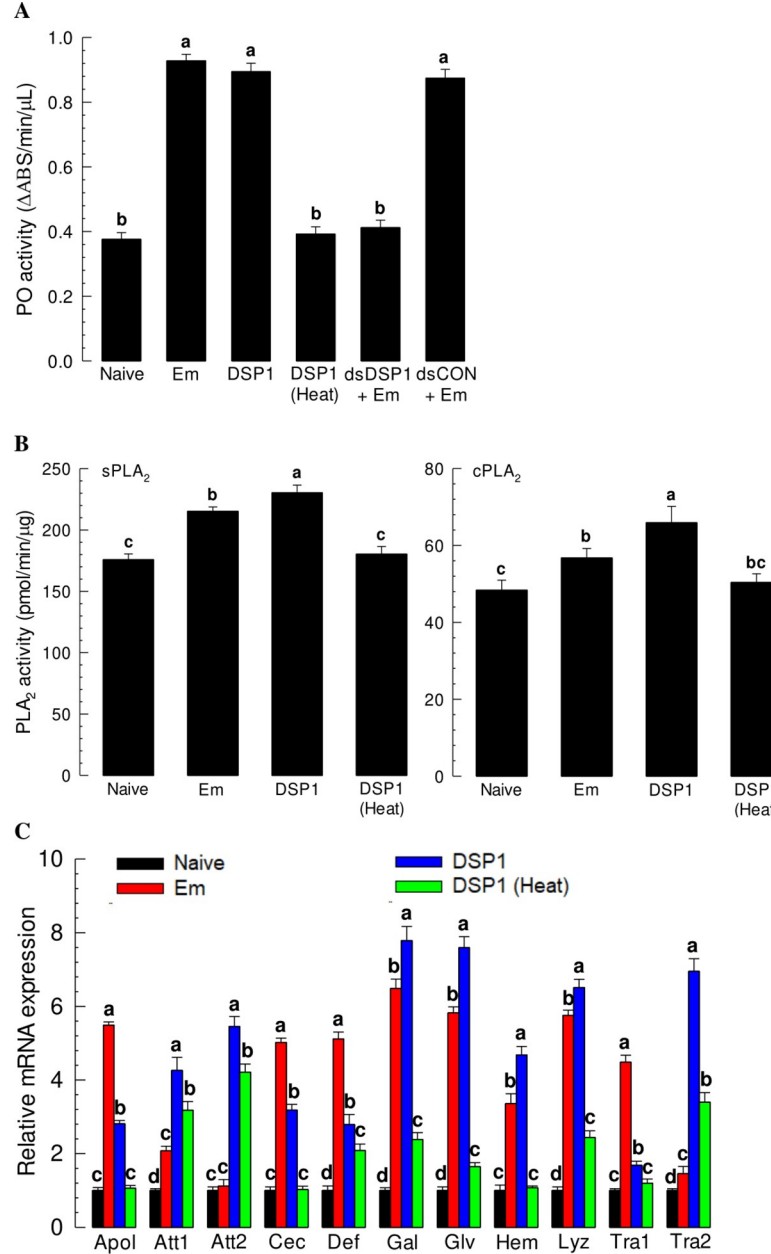

**Fig 2. Immune mediation of Se-DSP1 in *S. exigua*.** rSe-DSP1 ('DSP1') was injected into L5 larvae at a dose of 0.8 μg/larva. Inactivation of DSP1 used heat treatment at 95˚C for 10 min. Immune challenge used an injection of *E. mundtii* ('Em') to L5 larvae at a dose of $4 \times 10^5$ cells/larva. At 8 h PI, hemolymph and fat body were collected. Hemolymph was used for analyzing activities of phenoloxidase ('PO') and sPLA$_2$. Fat body was used for analyzing cPLA$_2$ activity. For antimicrobial peptide ('AMP') analysis, fat body from treated larvae was collected at 12 h PI. (A) Up-regulation of PO activity by Se-DSP1. RNAi used injection of gene-specific dsRNA against Se-DSP1 (dsDSP1) at a dose of 1 μg/larva. At 24 h PI, Em was used for treatment. Control dsRNA (dsCON) used dsRNA specific to a viral gene, *CpBV302*. (B) Up-regulation of PLA$_2$ activity by Se-DSP1. (C) Up-regulation of AMP gene expression. Expression levels of AMP were presented as fold changes in comparison with those in naïve larvae. Expression level of a ribosomal gene, *RL32*, was used as reference to normalize expression levels of target genes. Each treatment was replicated three times. Different letters above standard deviation bars denote significant difference among means at Type I error = 0.05 (LSD test).

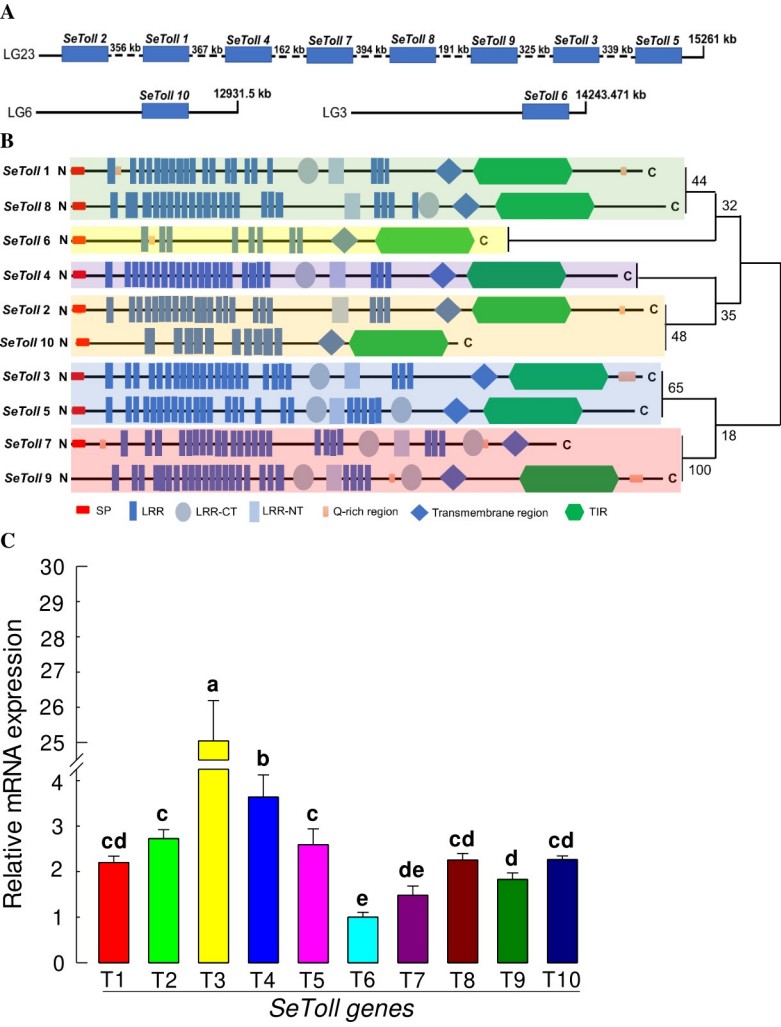

**Fig 3. Toll receptors (Se-Tolls) of *S. exigua*.** (A) Gene map of 10 *Se-Toll*s predicted in this study on chromosome(s), showing three linkage groups (LGs). (B) Their domain and phylogeny analyses. Predicted domains included SP (signal peptide), LRR (leucine-rich-repeat), LRR-CT (leucine-rich-repeat C terminal), LRR-NT (leucine-rich-repeat N terminal), Q-rich region (glutamine rich region), transmembrane region, and TIR (Toll-interleukin receptor). Neighbor-joining method was applied for constructing a phylogenetic tree. Bootstrap values on nodes are obtained from 1,000 repetitions. (C) Expression profile of 10 Toll genes ('T1-T10') in naive fat body tissues of *S. exigua*. Expression level of a ribosomal gene, *RL32*, was used as reference to normalize expression levels of target genes. Each treatment was replicated three times. Different letters above standard deviation bars denote significant difference among means at Type I error = 0.05 (LSD test).

Injection of individual dsRNA specific to each of 10 SeToll genes suppressed its target gene at 24 or 48 h after treatment compared to control dsRNA injection (Figs 4A and S2). However, some SeToll genes recovered their expression levels at 72 h after injection. Under these RNAi conditions, induction of PO activity in response to rSe-DSP1 injection was significantly suppressed compared to that after control RNAi treatment for three dsRNA treatments (against *SeToll5*, *SeToll6*, or *SeToll9* expression) (Fig 4B). PLA$_2$ activities in response to rSe-DSP1 injection were not induced by RNAi treatment against *SeToll1*, *SeToll2*, *SeToll5*, *SeToll6*, *SeToll7*, or *SeToll9* for sPLA$_2$, or against *SeToll2*, *SeToll3*, *SeToll4*, *SeToll5*, *SeToll6*, *SeToll7*, or *SeToll9* for cPLA$_2$ (Fig 4C). Induction of AMP gene expressions in response to rSe-DSP1 was also inhibited by RNAi specific to SeToll genes (Table 1). Especially, RNAi treatments against *SeToll2*

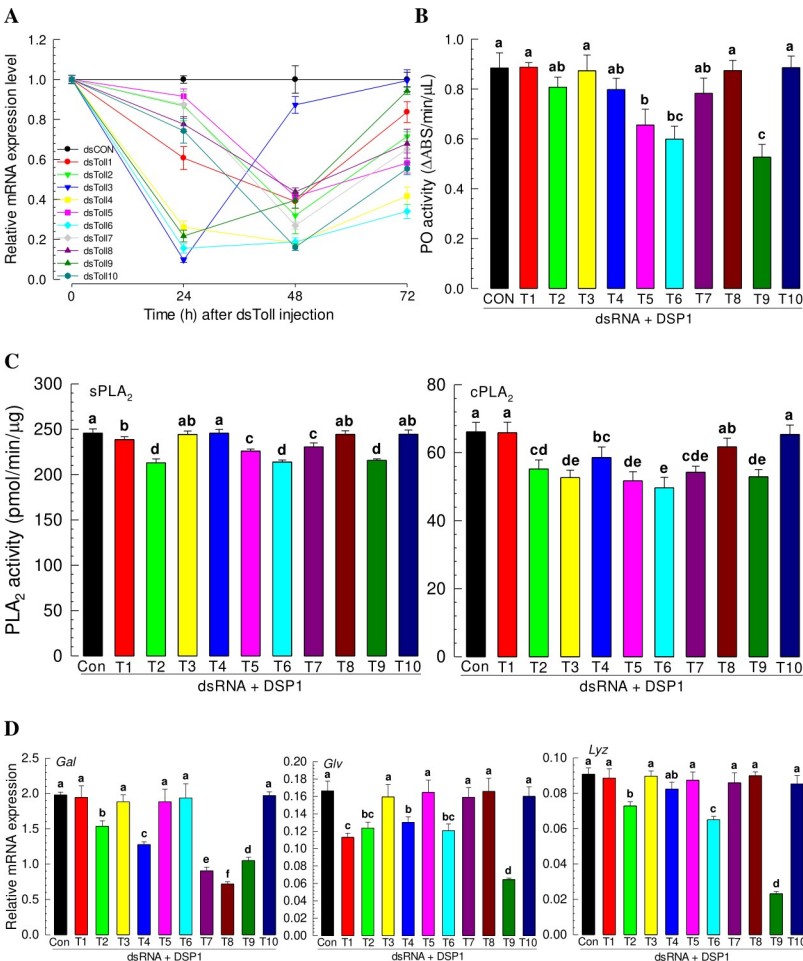

**Fig 4. Functional assay of 10 Se-Tolls ('T1-T10') for immune mediation of Se-DSP1 in *S. exigua* by individual knocking-down of gene expression using RNAi.** (A) RNAi efficiencies of 10 Se-Toll genes by injecting gene-specific dsRNAs ('dsToll1-dsToll10', 1 μg/larva) to L5 larvae. A viral gene, *CpBV302*, was used to prepare control dsRNA ('dsCON'). Expression level of a ribosomal gene, *RL32*, was used as reference to normalize expression levels of target genes. (B-D) Changes in immune responses after individual RNAi treatments. At 24 h PI dsRNA, rSe-DSP1 ('DSP1') was injected to L5 larvae at a dose of 0.8 μg/larva. At 8 h PI, hemolymph and fat body were collected. Hemolymph was used for analyzing activities of phenoloxidase ('PO') and sPLA$_2$. Fat body was used for analyzing cPLA$_2$ activity. For analysis of expression levels antimicrobial peptide (gallerimycin ('Gal'), gloverin ('Glv'), and lysozyme ('Lyz')) genes, fat bodies were collected from treated larvae at 12 h PI. Each treatment was replicated three times. Different letters above standard deviation bars denote significant difference among means at Type I error = 0.05 (LSD test).

and *SeToll9* expression significantly ($p < 0.05$) suppressed the induction of three AMPs (*Gal*, *Glv*, and *Lyz*), with RNAi against *SeToll9* being the most significant ($P < 0.05$) in suppressing the induction of these three AMP genes (Fig 4D). From these analyses, only RNAi treatment against *SeToll9* expression impaired all the immune responses mediated by Se-DSP1. These results suggest that SeToll9 plays a crucial role in mediating Se-DSP1 to induce immune responses in *S. exigua*.

Spz is a ligand for Toll receptor [22]. Two Spz genes were predicted from the genome of *S. exigua*. They shared a Toll-binding domain (Fig 5). Activation of proSpz to Spz requires proteolytic cleavage which releases the C-terminal clip domain (Fig 5A). These two Spz appeared to be distinct (Fig 5B). In response to bacterial challenge, *Se-Spz1* was not induced in hemocytes, although its expression level was significantly increased in fat bodies (Fig 5C). In

**Table 1. Effect of individual RNAi treatments against 10 Toll receptors of *S. exigua* on expression of antimicrobial peptide (AMP) genes.** Gene-specific dsRNA ('dsToll', 1 µg per larva) was injected into L5 larvae. After 24 h PI of dsRNA, rSe-DSP1 (0.8 µg per larva) was injected. After 12 h post injection of Se-DSP1, tissue samples (hemocyte ('HC'), fat body ('FB'), and midgut ('GT')) were collected and used for RT-qPCR.

| dsToll | Tissue | AMP genes[1] | | | | | | | | | | |
|--------|--------|------|------|------|-----|-----|-----|-----|-----|-----|------|------|
| | | Apol | Att1 | Att2 | Cec | Def | Gal | Glv | Hem | Lyz | Tra1 | Tra2 |
| Toll1 | HC | - | - | - | - | - | - | - | - | - | - | - |
| | FB | - | - | - | - | - | - | + | - | - | - | - |
| | GT | - | - | + | - | - | - | - | - | - | - | - |
| Toll2 | HC | + | - | + | - | + | + | + | + | - | + | + |
| | FB | - | + | + | - | - | + | + | - | + | - | - |
| | GT | + | - | + | + | + | - | - | + | - | + | + |
| Toll3 | HC | - | + | + | - | + | - | - | - | - | - | + |
| | FB | - | - | - | - | - | - | - | + | - | - | + |
| | GT | + | - | - | + | - | - | - | + | - | + | + |
| Toll4 | HC | - | - | + | - | - | - | + | - | + | - | - |
| | FB | - | - | + | + | + | + | + | - | - | - | + |
| | GT | + | + | + | + | + | + | + | + | + | + | + |
| Toll5 | HC | + | + | + | + | + | + | + | - | + | + | + |
| | FB | - | + | + | + | - | - | - | - | - | - | - |
| | GT | + | + | + | + | + | + | - | + | + | + | + |
| Toll6 | HC | + | - | + | + | + | + | + | - | + | + | + |
| | FB | - | + | + | + | - | - | + | + | + | - | + |
| | GT | + | + | + | + | + | + | + | + | + | + | + |
| Toll7 | HC | - | + | + | - | - | + | + | + | + | - | - |
| | FB | + | - | + | + | - | + | - | + | - | + | - |
| | GT | + | + | + | + | - | + | - | + | + | + | + |
| Toll8 | HC | - | - | - | - | + | - | - | - | - | - | - |
| | FB | + | - | - | + | - | + | - | - | - | + | - |
| | GT | - | - | - | - | + | - | - | + | - | - | - |
| Toll9 | HC | + | + | + | + | + | - | + | - | + | - | + |
| | FB | + | + | + | + | + | + | + | + | + | + | + |
| | GT | + | + | + | + | + | + | + | + | + | + | + |
| Toll10 | HC | + | - | - | - | - | + | + | + | - | + | - |
| | FB | - | - | + | + | - | + | + | - | - | - | + |
| | GT | + | + | + | - | + | + | - | + | - | + | + |

1 '+' represents significant ($P < 0.05$) decrease of AMP expression after Toll RNAi compared to control while '-' represents no change from control.

addition, rSe-DSP1 did not induce *Se-Spz1* in fat bodies while it significantly induced the two Spz genes in hemocytes. These suggest that both *Se-Spz* expressions might be not directly induced by the bacterial challenge or Se-DSP1 injection.

RNAi treatment against *Se-Spz1* or *Se-Spz2* suppressed their expression levels for at least 72 h PI (Figs 6A and S2). Under RNAi conditions, induction of PO activity in response to bacterial or rSe-DSP1 injection was not suppressed (Fig 6B). However, the induction of PLA$_2$ activity was significantly ($p < 0.05$) suppressed by RNAi treatment against either *Se-Spz1* or *Se-Spz2* (Fig 6C). Regarding AMP gene induction after injection of Se-DSP1, RNAi against *Se-Spz1* suppressed the induction of expression of three AMP genes (*Gal*, *Glv*, and *Lyz*) while RNAi against *Se-Spz2* suppressed the expression of *Glv* and *Lyz*, but not *Gal* (Fig 6D). These results suggest that Toll-Spz immune signalling pathway is mediated by Se-DSP1 in response to challenge by Gram-positive bacteria.

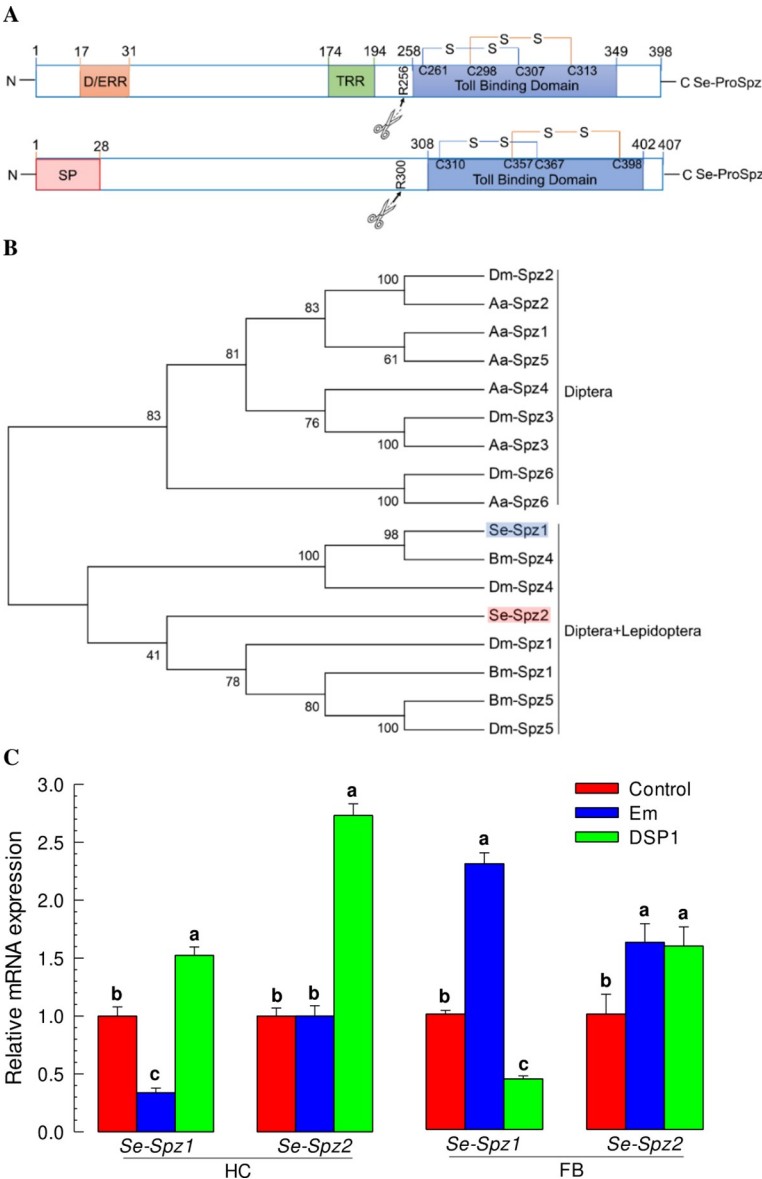

**Fig 5. Two Spätzles (Se-Spz) of *S. exigua*.** (A) Functional domains of two proSpätzles ('Se-ProSpz1' and 'Se-Spz2'), including a SP (signal peptide), a D/ERR (aspartic/glutamic acid rich region), a TRR (threonine rich region), and a Toll binding domain. Scissors indicate cleavage sites during post-translational modification from Se-ProSpz1/2 to active Se-Spz1/2. Disulfide bonds are indicated by linking cysteine residues. (B) Phylogeny analysis of Se-ProSpz1/2 and those of other insects. The analysis was performed using MEGA6 program with a Neighbor-joining method. Bootstrapping values were obtained with 1,000 repetitions to support branching and clustering. Amino acid sequences were retrieved from GenBank with accession numbers shown in S2 Table. (C) Expression profile of Se-Spz1/2 in hemocyte ('HC') and fat body ('FB') of L5 larvae of *S. exigua*. rSe-DSP1 ('DSP1') was injected into L5 larvae at a dose of 0.8 μg/larva. Immune challenge was performed by injecting *E. mundtii* ('Em') to L5 larvae at a dose of $4 \times 10^5$ cells/larva. At 8 h PI, hemolymph and fat body were collected for RT-qPCR analysis. Control insects were injected with PBS. Expression levels of *Se-Spz1/2* were presented as fold changes in comparison with those in control larvae. Expression level of a ribosomal gene, *RL32*, was used as reference to normalize expression levels of target genes. Each treatment was replicated three times. Different letters above standard deviation bars denote significant difference among means at Type I error = 0.05 (LSD test).

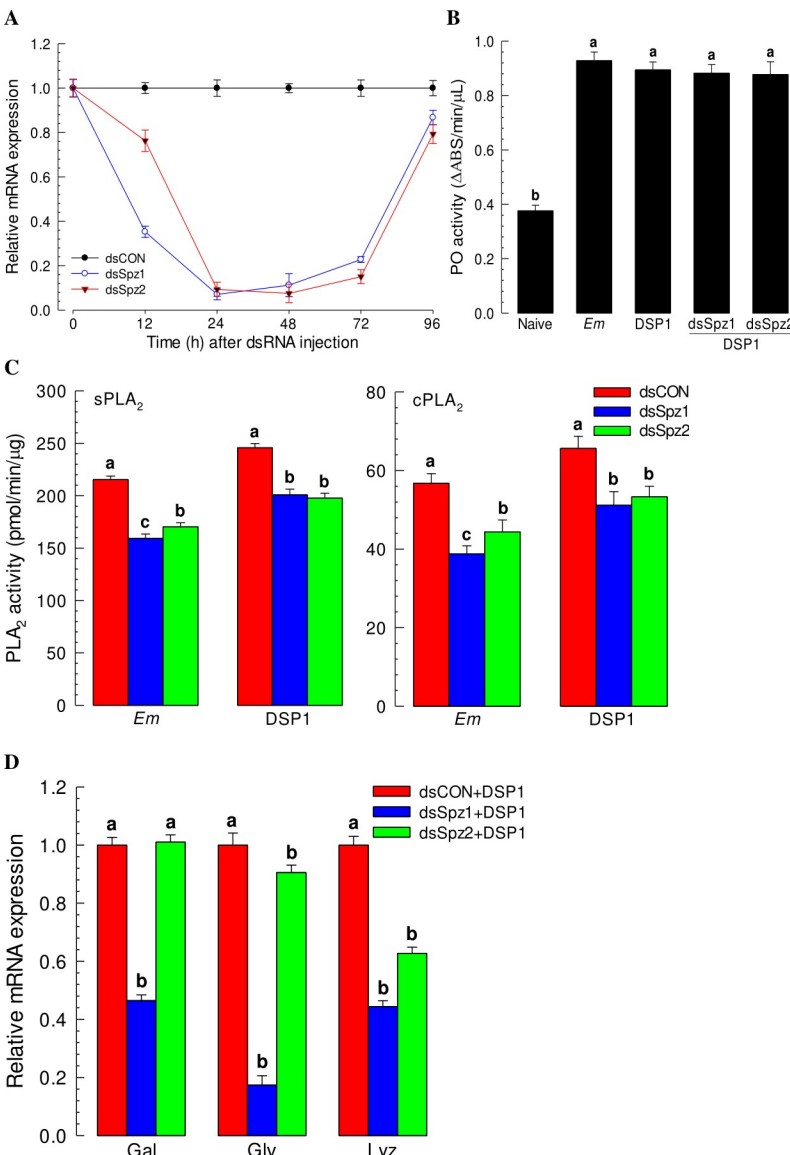

**Fig 6. Functional assay of two Spätzles ('Se-Spz1 and Se-Spz2') for immune mediation of Se-DSP1 in *S. exigua* by individual knocking-down of gene expression using RNAi.** (A) RNAi efficiencies of two Se-Spz genes by injecting gene-specific dsRNAs ('dsSpz1 and dsSpz2', 1 µg/larva) to L5 larvae. A viral gene, *CpBV302*, was used to prepare control dsRNA ('dsCON'). Expression level of a ribosomal gene, *RL32*, was used as reference to normalize expression levels of target genes. (B-D) Changes in immune responses after individual RNAi treatment. At 24 h PI dsRNA, rSe-DSP1 ('DSP1') was injected to L5 larvae at a dose of 0.8 µg/larva. For bacterial immune challenge, *Enterococcus mundtii* (Em, $4 \times 10^5$ cells/larva) was injected into larvae. Naïve larvae were injected with PBS. At 8 h PI, hemolymph and fat body were collected. Hemolymph was used for analyzing activities of phenoloxidase ('PO') and sPLA$_2$. Fat body was used for analyzing cPLA$_2$ activity. For analyzing expression levels of antimicrobial peptide (gallerimycin ('Gal'), gloverin ('Glv'), and lysozyme ('Lyz')) genes, fat bodies were collected from treated larvae at 12 h PI. Each treatment was replicated three times. Different letters above standard deviation bars denote significant difference among means at Type I error = 0.05 (LSD test).

## SeToll9 deletion mutants of *S. exigua* lose responsiveness to Se-DSP1 injection

We performed CRISPR/Cas9-mediated mutagenesis of *Se-DSP1* (Fig 7). *Se-DSP1* has an exon without intron. Based on this, we designed two sgRNA sites (covering 362 bp fragment,

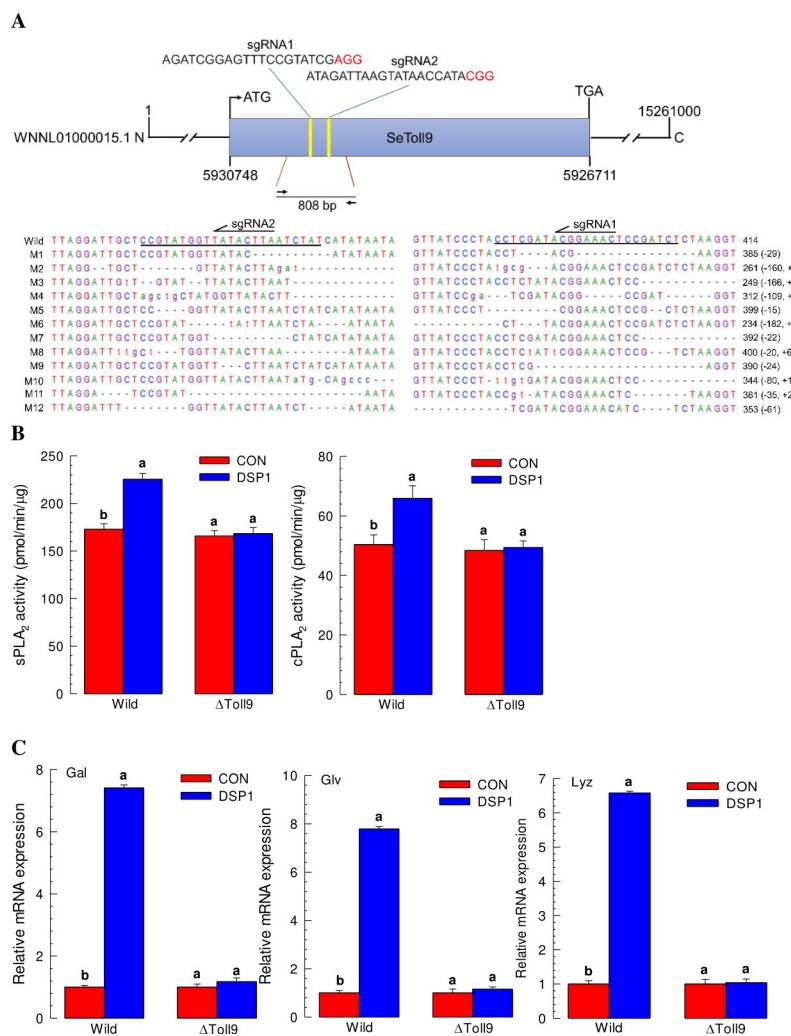

**Fig 7. Deletion mutants of *SeToll9* using CRISPR/Cas9 and their insensitivity to Se-DSP1 to express immune responses in *S. exigua*.** (A) Construction of *SeToll9*-deletion mutants (ΔToll9) with two single-stranded guide RNAs (sgRNAs), with protospacer adjacent motif (PAM) denoted in red color. Twelve different types of ΔToll9 ('M1-M12') are confirmed by sequence analysis of 808 bp around both sgRNA-specific deletion sites by comparing with the corresponding sequence of wild type ('Wild'). Deletion or insertion sizes by CRISPR/Cas9 are denoted by '+' or '-' in parentheses. (B) Insensitivity of mutants to Se-DSP1 in activation of PLA$_2$. rSe-DSP1 ('DSP1') was injected into L5 larvae at a dose of 0.8 μg/larva. Control ('CON') larvae were injected with PBS. At 8 h PI, hemolymph and fat bodies were collected. Hemolymph was used for sPLA$_2$ activity analysis. Fat body was used for cPLA$_2$ activity analysis. (C) Insensitivity of mutants to Se-DSP1 in activating expression of antimicrobial peptide (gallerimycin ('Gal'), gloverin ('Glv'), and lysozyme ('Lyz')) genes. Fat bodies were collected from treated larvae at 12 h PI of Se-DSP1. Expression level of a ribosomal gene, *RL32*, was used as reference to normalize expression levels of target genes. Each treatment was replicated three times. Different letters above standard deviation bars denote significant difference among means at Type I error = 0.05 (LSD test).

Fig 7A). A mixture of sgRNAs and Cas9 was injected into newly-deposited eggs. After eggs were injected with normal food dye, the hatching rate decreased from normally > 90% to 54.9%. Injecting the CRISPR/Cas9 construct reduced hatching rate to 11.3%. We then generated 12 mutants and confirmed by sequencing gDNA. Deletion sizes of these mutants ranged from 15 to 182 bp. Insertions between two sgRNA sites were also observed. Knocking out *Se-DSP1* impaired the responsiveness of *S. exigua* to DSP1 injection. Injection of DSP1 to these

mutants did not induce $PLA_2$ activities in these mutants, although such injection induced $PLA_2$ activities in the wild type (Fig 7B). Injection of DSP1 did not up-regulate the expression of AMP genes in these mutants either compared to the wild type larvae (Fig 7C).

### Inhibition of Se-DSP1 immune signalling enhances Bt pathogenicity

Toll-Spz immune signalling pathway mediated by Se-DSP1 may play a crucial role in defending Gram-positive bacteria. To test this hypothesis, *B. thuringiensis* (Bt), a well-known Gram-positive insect pathogen, was applied to *S. exigua* larvae after individual RNAi treatment against each signaling component (Fig 8). At 24 h after indicidual dsRNA injections against ten Se-Toll genes, 500 ppm of Bt was applied to larvae by feeding with diet (Fig 8A). Compared to control RNAi treatment, four RNAi treatments against *SeToll5*, *SeToll6*, *SeToll7*, and *SeToll9* significantly ($p < 0.05$) enhanced Bt pathogenicity. Treatment with RNAi specific to *SeToll9* expression was the most effective one in enhancing the insecticidal activity of Bt. Similarly, two *Se-Spz* genes were tested. Results showed that both *Se-Spz* genes should be expressed to defend against Bt pathogenicity (Fig 8B).

### *X. hominickii* secretes secondary metabolites that inhibit DSP1 translocation

To look for effective inhibitor(s) of Se-DSP1, bacterial culture broth of *X. hominickii*, an insect pathogen known to induce immunosuppression [7], was fractionated (S4 Fig) and the resulting fractions were assessed for their binding affinities for rSe-DSP1 (Fig 9A). Four organic extracts were assessed. Of them, the butanol extract ('BX') was the most effective one in binding to Se-DSP1 with the lowest Kd value (Fig 9B). BX was further fractionated into 15 subfractions (Fig 9C). Three subfractions (F2, F4, and F6) showed the lowest Kd values. Thus, they were subjected to further fractionation. From F2 subfraction, nine subfractions were obtained and three subfractions (F2-2, F2-5, and F2-9) were selected after performing rSe-DSP1 binding assays. From F4 fractions, 11 subfractions were obtained and two subfractions (F4-3 and F4-8) were selected after rSe-DSP1 binding assays. From F6 fractions, nine subfractions were obtained and three subfractions (F6-3, F6-4, and F6-8) were selected after rSe-DSP1 binding assays. These eight subfractions selected were then subjected to GC-MS (S5 Fig).

Seven metabolites were predicted from active subfractions using GC-MS (Fig 10A). When these compounds were tested to bind to Se-DSP1, they showed binding affinities at a low micromolar range (Fig 10B). Especially, 3-ethoxy-4-methoxyphenol (EMP) exhibited the highest binding affinity. Although its chemical structure was similar to salicylic acid (SA), EMP had higher binding affinity to rSe-DSP1 than SA known to bind to rSe-DSP1 and inhibit immune responses of *S. exigua* [11].

### EMP induces immunosuppression and enhances Bt pathogenicity

The high binding affinity of EMP to rSe-DSP1 was further analyzed for its effect on immune responses mediated by Se-DSP1. EMP treatment appeared to inhibit rSe-DSP1 release from damage fat bodies (Fig 11A) and hemocytes (Fig 11B). Se-DSP1 was localized in nuclei of hemocytes and fat bodies of naïve larvae. Upon bacterial challenge, Se-DSP1 was released into plasma (Fig 11C), However, EMP treatment prevented the release of Se-DSP1 after bacterial challenge.

The suppression of Se-DSP1 release by treatment with EMP also significantly suppressed the induction of PO activity after Se-DSP1 injection (Fig 12A). This suppression of Se-DSP1 activity was also observed after SA treatment. EMP and SA both suppressed $PLA_2$ activity

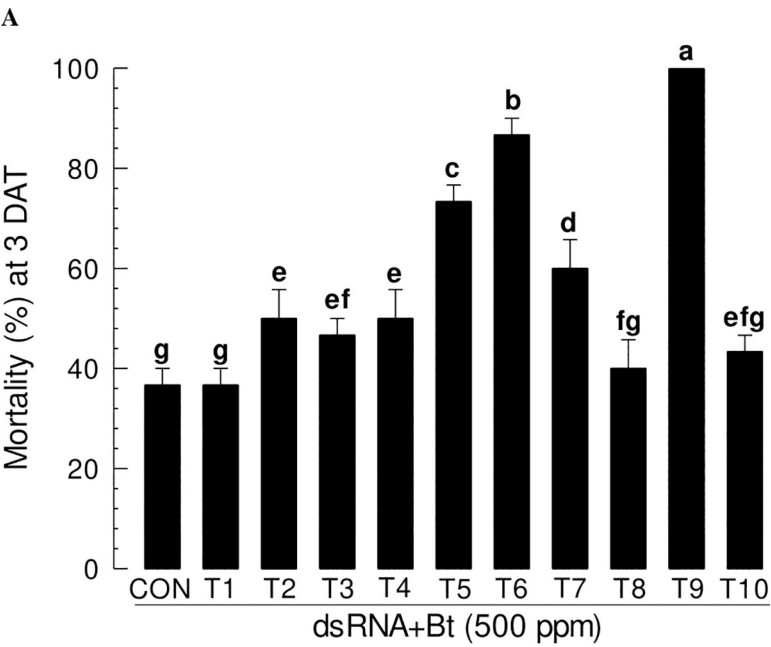

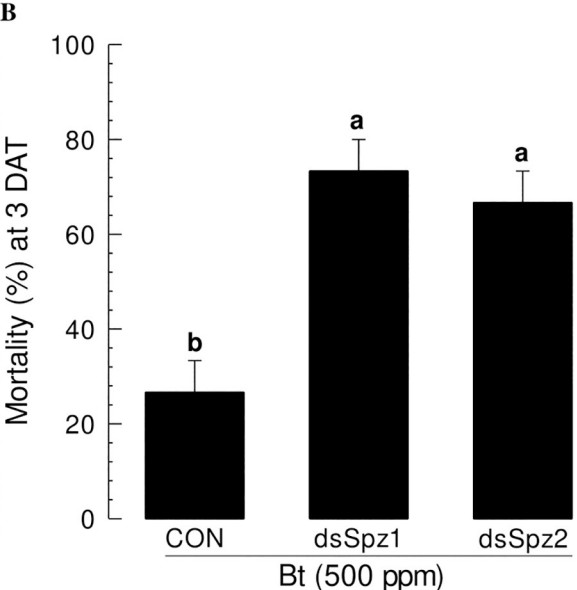

**Fig 8. Comparative analysis of susceptibility of *S. exigua* larvae to an entomopathogenic bacterium, *Bacillus thuringiensis* ('Bt'), after individual RNAi treatment targeting each of 10 SeTolls ('T1-T10') (A) and two Spätzle ('SeSpz1 and SeSpz2') (B) genes.** dsRNA-specific to individual genes were injected to L5 larvae (1 μg/larva). Control larvae were injected with dsRNA ('CON') specific to a viral gene *CpBV302*. At 24 h PI, 500 ppm of Bt was orally fed to larvae with a leaf-dipping method. Mortality was recorded at 3 days after treatment ('DAT'). Each treatment was replicated three times and each replication used 10 larvae. Different letters above standard deviation bars denote significant difference among means at Type I error = 0.05 (LSD test).

induction after Se-DSP1 treatment, with EMP being more potent than SA (Fig 12B). The induction of expression of three AMP genes was significantly suppressed by treatment with EMP or SA (Fig 12C). Sch immunosuppressive activity of EMP significantly enhanced Bt insecticidal activity (Fig 12D).

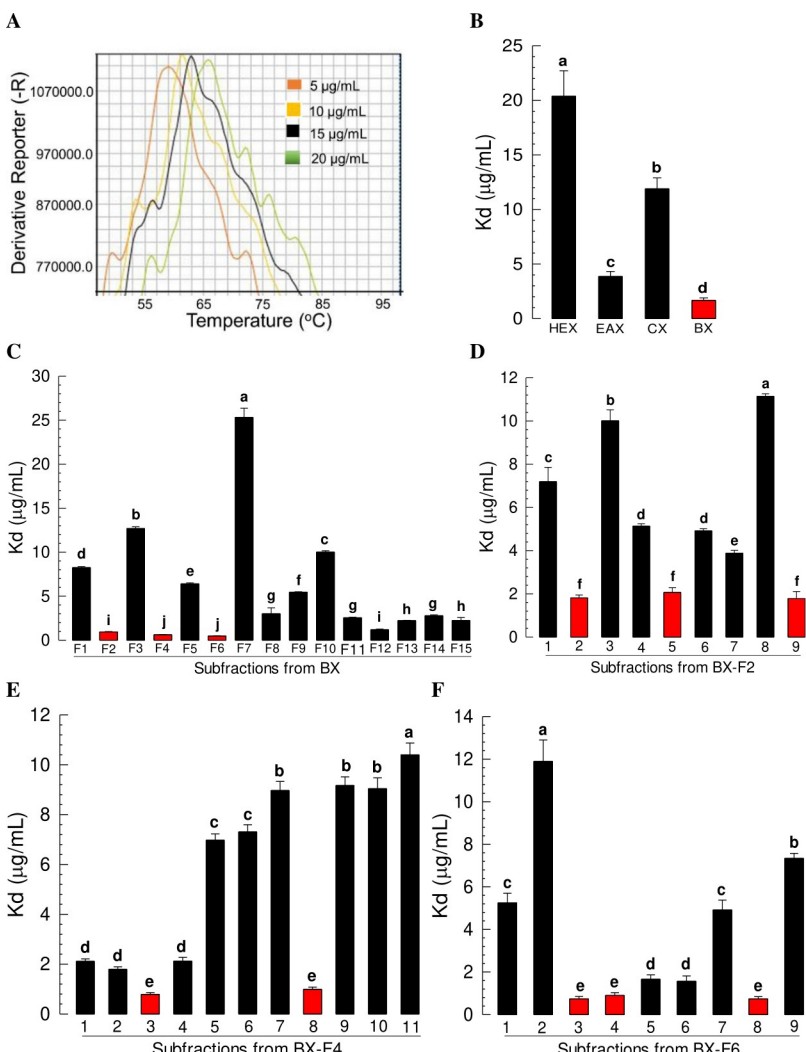

**Fig 9. Screening bacterial metabolites of *X. hominickii* (Xh) for their binding affinities for Se-DSP1.** (A) Thermal shift assay for screening binding affinities using protein denaturation curve occurring with increasing ambient temperature. Shifting of the maximal value at higher temperature presumes that Se-DSP1 is completely denatured and maximally bound to a fluorescence dye. Dissociation constant (Kd) is estimated based on the relation between the maximal dissociation temperature and test compound concentration. (B) Binding assays for four organic extracts of Xh culture broth after 48 h of grown in TSB (S2 Fig). Extracts used included hexane (HEX), ethyl acetate (EAX), chloroform (CX), and butanol (BX) extracts. (C) Binding assay of 15 fractions ('F1-F15') of BX. (D) Binding assays for nine subfractions isolated from F2 fraction of BX ('BX-F2'). (E) Binding assays for 11 subfractions isolated from F4 fraction of BX ('BX-F4') (F) Binding assays for nine subfractions isolated from F6 of BX ('BX-F6'). Each measurement was replicated three times (three independent samples for each replicate). Different letters above standard deviation bars indicate significant differences among means at Type I error = 0.05 (LSD test).

## Discussion

Upon infection or physical damage, tissues and specific cells can release damage signals via DAMP [14]. Se-DSP1 was the first DAMP molecule reported in insects [11]. The present study further investigated the release of Se-DSP1 and its immune signalling via Toll-Spz pathway. *X. hominickii*, a well-known insect pathogen, can induce insect immunosuppression by inhibiting eicosanoid biosynthesis [23]. This study tested a hypothesis that *X. hominickii* inhibits the DAMP to induce host immunosuppression.

**A**

| Xh-BX | MW | Compounds | Structure |
|---|---|---|---|
| F2-2 | 168.079 | 3-Ethoxy-4-methoxyphenol | |
| F2-5 | 210.137 | Hexahydro-3-(2-methylpropyl)-pyrrolopyrazine-1,4-dione | |
| F2-9 | 390.277 | Bis(2-ethylhexyl) phthalate | |
| F4-3 | 117.15 | Indole | |
| F4-8/F6-4 | 129.152 | Dibutylamine | |
| F6-3 | 147.032 | Phthalimide | |
| F6-8 | 147.032 | o-Cyanobenzoic acid | |
| | 138.121 | Salicylic acid | |

**B**

| Compounds | Kd ($\mu$M) ± SD |
|---|---|
| 3-Ethoxy-4-methoxyphenol | 0.119 ± 0.02 c |
| Hexahydro-3-(2-methylpropyl) pyrrolopyrazine-1,4-dione | 0.155 ± 0.04 bc |
| Bis(2-ethylhexyl) phthalate | 0.211 ± 0.03 b |
| Indole | 0.145 ± 0.03 bc |
| Dibutylamine | 0.379 ± 0.07 a |
| Phthalimide | 0.142 ± 0.02 bc |
| o-Cyanobenzoic acid | 0.166 ± 0.02 bc |
| Salicylic acid | 0.238 ± 0.04 b |

**Fig 10. Identification of compounds binding to Se-DSP1 from bacterial culture broth of *X. hominickii*.** (A) Prediction of Se-DSP1-binding compounds from purified butanol extract ('Xh-BX') of *X. hominickii* culture broth using GC-MS analysis. For example, 'F2-2' stands for subfraction #2 from 'BX-F2' fraction in Fig 9. See GC-MS chromatograms of compounds in S3 Fig (B) Binding affinity (Kd) estimations of seven bacterial metabolites and salicylic acid to rSe-DSP1 assessed by thermal shift assay. Each treatment was replicated three times with individual samples. Different letters following standard deviation ('SD') indicate significant difference among means at Type I error = 0.05 (LSD test).

Se-DSP1 in the nucleus was secreted to the plasma in a form of exosome in infected larvae of *S. exigua*. Se-DSP1 is a homologous protein of a vertebrate HMGB1 [11]. HMGB1 is a ubiquitously expressed and highly conserved nuclear protein that plays important roles in chromatin organization and transcriptional regulation [12]. This nuclear HMGB1 is released to the plasma under stress either passively from dead cells or actively by secretion from activated immune cells, enterocytes, hepatocytes, and possibly several other types of cells [13]. Released HMGB1 can act as a DAMP and activate the innate immune system by interacting with

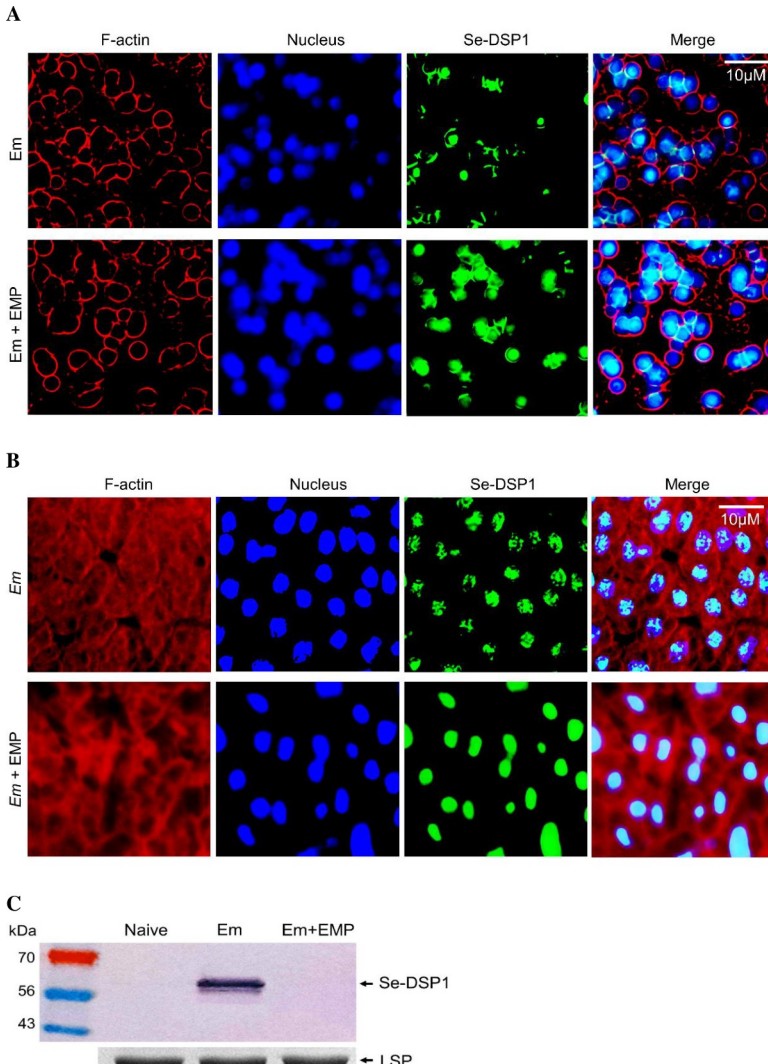

**Fig 11. Inhibitory activity of 3-ethoxy-4-methoxyphenol ('EMP') on Se-DSP1 secretion to plasma in response of *S. exigua* to immune challenge.** L5 larvae were injected with *E. mundtii* ('Em', $4 \times 10^5$ cells/larva) and EMP (1 μg/larva). At 6 h PI, hemolymph and fat body were collected. Immunofluorescence assays of Se-DSP1 in hemocytes (A) and fat body (B) are shown. F-actin and nucleus were stained with phalloidin and DAPI, respectively. Se-DSP1 was detected with its polyclonal antibody. (C) Western blotting analysis of Se-DSP1 in the plasma of naïve or treated larvae. Each lane was loaded with 5 μL of plasma. Coomassie-staining bands against a larval storage protein ('LSP') indicated that the same amount of proteins in plasma samples was loaded.

pattern recognition receptors [14]. It has been demonstrated that several receptors are involved in HMGB1-mediated functions, including RAGE (receptor for advanced glycation end products), TLR2 (Toll-like receptor 2), TLR4, and TLR9 [18,24]. Therefore, Se-DSP1 has been regarded as a DAMP molecule and a key player in activating immune responses in insects [11]. However, how Se-DSP1 is released from the nucleus to the extracellular plasma remains unclear. In mammals, HMGB1 is released from the nucleus using a caspase pathway and delivered to the plasma via exosomes. This has been well demonstrated in hepatocytes in response to lipopolysaccharide via Toll-like receptor 4/caspase-11-dependent cleavage [25]. The present study observed that Se-DSP1 was located in the exosome collected from larval plasma of *S. exigua*. Exosomes are membrane-bound extracellular vesicles produced in the endosomal

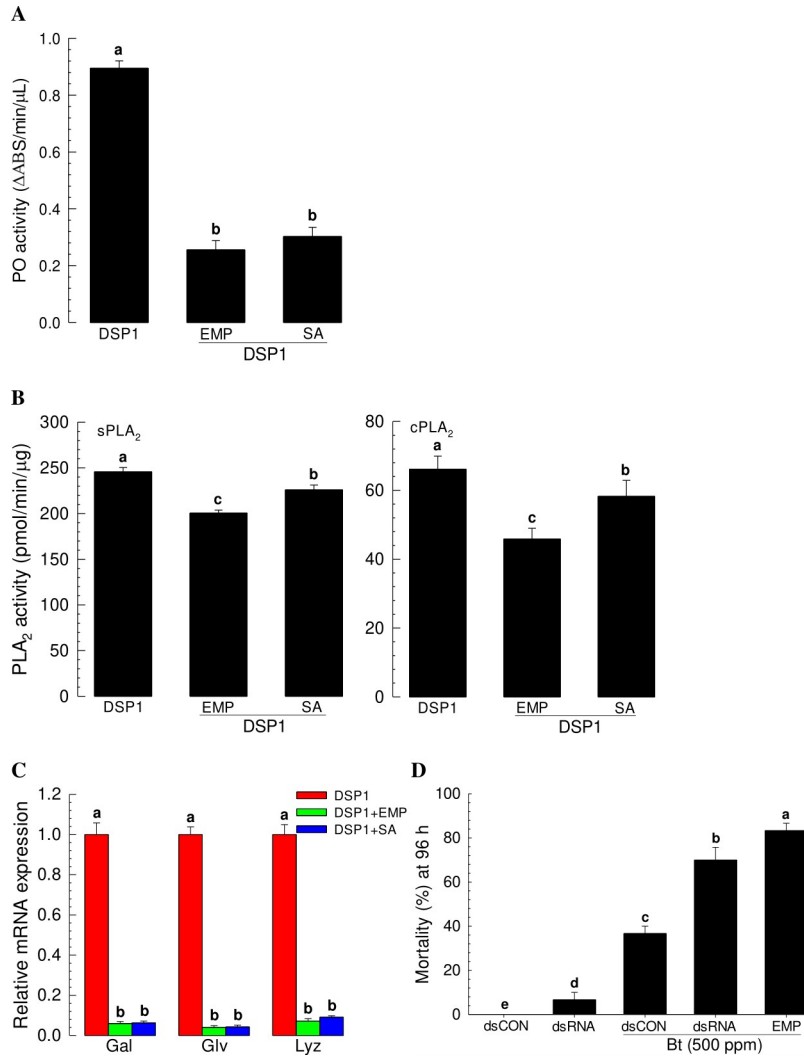

**Fig 12. Inhibitory effect of 3-ethoxy-4-methoxyphenol ('EMP') or salicylic acid ('SA') on immune responses mediated by Se-DSP1 in *S. exigua*.** L5 larvae were injected with rSe-DSP1 ('DSP1', 0.8 μg/larva) and EMP or SA (1 μg/larva). At 8 h PI, hemolymph and fat body were collected. Hemolymph was used for analyzing activities of phenoloxidase ('PO') and sPLA$_2$. Fat body was used for analysis of cPLA$_2$ activity. For antimicrobial peptide ('AMP') analysis, fat bodies were collected from treated larvae at 12 h PI. (A) Inhibition of up-regulated PO activity by EMP or SA. (B) Inhibition of up-regulated PLA$_2$ activities by EMP or SA. (C) Inhibition of up-regulation of antimicrobial peptide (gallerimycin ('Gal'), gloverin ('Glv'), and lysozyme ('Lyz')) genes by EMP or SA. Expression levels of AMP were presented as fold changes in comparison with those in naïve larvae. Expression level of a ribosomal gene, *RL32*, was used as reference to normalize expression levels of targeted genes. Each treatment was replicated three times. Different letters above standard deviation bars denote significant difference among means at Type I error = 0.05 (LSD test). (D) Enhanced susceptibility of *S. exigua* larvae to *Bacillus thuringiensis* ('Bt'), an entomopathogenic bacterium, after treatment with RNAi specific to Se-DSP1 or EMP. RNAi was performed by injecting gene-specific dsRNA against Se-DSP1 (dsDSP1) at a dose of 1 μg/larva. At 24 h PI, Bt was used for treatnet. Control dsRNA (dsCON) used dsRNA specific to a viral gene, *CpBV302*. EMP treatment was performed by mixing with Bt suspension. Each treatment was replicated three times. Different letters above standard deviation bars denote significant difference among means at Type I error = 0.05 (LSD test).

compartment of most eukaryotic cells. They have been detected in biological fluids including blood, urine, and cerebrospinal fluid, with a diameter of at about 30 ~ 150 nm [26]. Exosomes can transfer molecules from one cell to another via membrane vesicle trafficking, thereby playing a crucial role in communicating immune signalling between cells [27]. In insects,

exosome-like vesicles have been found in hypopharyngeal gland secretomal products (honey, royal jelly, and bee pollen) of honeybee, *Apis mellifera*. They are known to possess antibacterial and pro-regenerative effects [28]. In *Drosophila*, exosomes play a direct immunological role in defending viral infection by delivering short interfering double stranded RNA derived from infected hemocytes [29]. In our current study, isolated exosomes containing Se-DSP1 reacted with antisera raised against CD9, a tetraspanin protein known to be rich in exosomes [30]. These results suggest that Se-DSP1 is released from the nucleus upon bacterial infection and secreted to the plasma in a form of exosome.

The Se-DSP1 secreted in plasma used the Toll/Spz signalling pathway to mediate immune responses. PO activity was significantly increased after bacterial challenge. However, RNAi against *Se-DSP1* expression prevented PO activation against bacterial challenge. In contrast, Se-DSP1 alone without bacterial challenge significantly increased the activity of PO. In response to Gram-positive bacterial challenge, *S. exigua* larvae express Toll immune signalling [17]. In insects, pathogen recognition accompanies the activation of a group of serine proteases with multiple modular regulatory domains [31] and proteases with amino-terminal clip domains [32]. Terminal clip proteases in the PO-activation pathway can activate PO which catalyzes the formation of reactive compounds and melanin to kill and sequester pathogens [33]. This suggests that Se-DSP1 may interact with the serine protease cascade to up-regulate PO activity. The activation of serine protease(s) by Se-DSP1 also cleaves pro-Spz to active Spz, which in turn binds to the Toll receptor to induce synthesis of AMPs [34]. The present study revealed that two Spz genes annotated were expressed in *S. exigua*. It is known that *D. melanogaster* genome encodes six Spz homologs [35]. All these six orthologs have been identified in genomes of mosquitoes *Anopheles gambiae* [36] and *Aedes aegypti* [37]. However, only two *Spz* homologs are present in the genome of honeybee *Apis mellifera* [38] and the genome of red flour beetle *Tribolium castaneum* [39]. Spz can activate immune responses via Toll signal pathway in insects. RNA interference experiments have demonstrated that *A. aegypti* Spz1 has a function in antifungal immunity [37]. Injection of the active form of *B. mori* Spz1 protein can induce antimicrobial peptide expression [40]. RNAi treatments against two Spz genes of *S. exigua* significantly interfered with immunity modulation of Se-DSP1 in the present study.

A total of 10 Toll receptor homologs in *S. exigua* were predicted. Although all Toll genes were expressed and associated with AMP expression, SeToll9 was found to have the strongest association with the Se-DSP1 signalling pathway based on individual RNAi treatment with Se-DSP1 injection. Toll immune signalling pathway is activated when Spz binds to Toll receptor [20]. Binding of Spz to Toll receptor promotes Toll multimerization which stimulates down-stream signalling through adaptor protein MyD88 and two kinases, Tube and Pelle [41]. This results in phosphorylation and degradation of IκB, Cactus, which releases NF-κB transcriptional factors Dif and Dorsal to translocate to the nucleus and induce AMP gene expression [42]. In *B. mori*, 14 Toll genes are identified, of which six genes are predicted to be associated with immunity based on a phylogenetic analysis [43]. So far, only BmToll9-1 has been confirmed to have function in innate immunity [44,45]. Shafeeq et al. [17] have shown that Pelle can activate $PLA_2$ and lead to eicosanoid biosynthesis. Although $PLA_2$ activity is required in various physiological processes, it is essential to mediate various immune responses in insects [4,6]. Eicosanoids can mediate various immune responses, including PO activation by releasing inactive proPO to the plasma [4]. These results indicate that Se-DSP1 can activates immune response via the Toll/Spz immune signalling pathway. This was further supported by findings that RNAi treatment against SeToll or Spz expression made larvae become susceptible to Bt pathogens.

The role of Se-DSP1 in mediating various immune responses of *S. exigua* suggests that microbial pathogens can inhibit Se-DSP1 to induce host immunosuppression for their survival

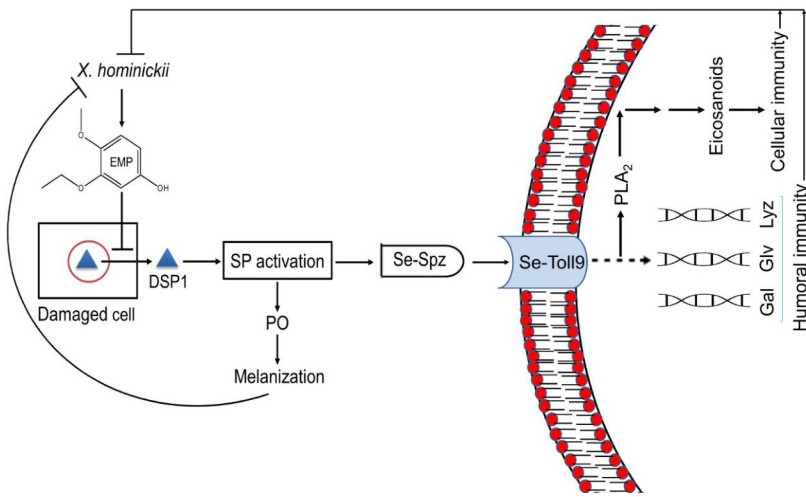

**Fig 13. A working hypothesis of Se-DSP1 for mediating immune responses via Toll-Spätzle ('Spz') signalling pathway and its inhibition by a bacterial metabolite, 3-ethoxy-4-methoxyphenol ('EMP'), in *S. exigua*.** Upon challenge with bacteria including an entomopathogenic bacterium, *X. hominickii*, Se-DSP1 is secreted to the plasma to activate serine protease (SP) cascade for activating phenoloxidase ('PO') and Spz. Activated PO can catalyze melanin formation to suppress the growth of pathogenic bacteria. Activated Spz can bind to SeToll receptor to activate PLA$_2$ and the expression of antimicrobial peptide (gallerimycin ('Gal'), gloverin ('Glv'), and lysozyme ('Lyz')) genes. Activated PLA$_2$ can catalyze eicosanoid biosynthesis to mediate cellular immune responses to defend bacterial infection along with AMPs. To overcome tmmune responses mediated by Se-DSP1, *X. hominickii* produces and secretes secondary metabolites including EMP to inhibit the secretion of Se-DSP1 and prevent its immune-mediating activity.

and growth. To test this hypothesis, this study screened secondary metabolites of *X. hominickii* after organic extraction. A previous study has shown that organic extract is more potent in inhibiting immune responses of *S. exigua* than an aqueous extract of secondary metabolities of another pathogen, *Xenorhabdus nematophila* [46]. Potent metabolites include phthalimide, indole, and salicylic acid (SA) derivatives. Especially, EMP, a SA derivative, was the most potent in binding to Se-DSP1. Treatment with EMP alone induced significant immunosuppression. Interestingly, Se-DSP1 was not secreted to the plasma in EMP-treated larvae. This suggests that EMP can bind to Se-DSP1 and prevent its translocation from the nucleus. Pathogenic bacteria including *X. hominickii* possess a gene cluster (polyketide ketone synthase: PKS) associated with the biosynthesis of virulent factors [47]. EMP is likely to be a product of PKS [48].

In summary, *X. hominickii*, an insect pathogen, can synthesize and secrete secondary metabolites including EMP (Fig 13). EMP then enters immune-associated cells to prevent secretion of Se-DSP1 for mediating immune responses via the Toll/Spz signalling pathway.

## Materials and methods

### Insect rearing and bacteria culture

Larvae of *S. exigua* were collected from Welsh onion (*Allium fistulsum* L.) fields in Andong, Korea and reared on an artificial diet [49]. Larvae underwent five larval molts (L1-L5). Adults were fed with 10% sucrose solution. Rearing conditions were 25 ± 2°C with relative humidity at 60 ± 5% and 16 h of day light length. *X. hominickii* was isolated from *S. monticolum* and grown in tryptic soy broth (TSB: Difco, Sparks, MD, USA) at 28°C for 48 h [1]. *Enterococcus mundtii* (*Em*), a Gram-positive bacterium, was also cultured overnight in TSB medium at 28°C with shaking at 180 rpm. For immune challenge, *E. mundtii* was centrifuged at

10,000 × g for 5 min. The cell pellet was dissolved in sterile distilled water and heat-killed at 90˚C for 10 min. Heat-killed *E. mundtii* was then injected into L5 larvae (4 x $10^5$ cells/larva) with a microsyringe (Hamilton, Reno, NV, USA) after counting with a hemocytometer (Neubauer improved bright line, Superior Marienfeld, Lauda-Konigshofen, Germany) under a phase contrast microscope (BX41, Olympus, Tokyo, Japan).

## Chemicals

Phthalimide (PM), 3-Ethoxy-4-methoxyphenol (EMP), o-cyanobenzoic acid (CBA), dibutylamine (DBA), bis-2 (ethylhexyl) phthalate (BEP), indole (IND), hexahydro-3-(2-methylpropyl)-pyrrolopyrazine-1,4-dione (HMPP), and salicylic acid (SA) were purchased from Sigma-Aldrich Korea (Seoul, Korea) and dissolved in dimethylsulfoxide (DMSO). Phosphate-buffered saline (PBS, pH 7.4) was prepared with 100 mM phosphoric acid and 0.7% sodium chloride. Anticoagulant buffer (ACB, pH 4.5) was prepared to contain 186 mM NaCl, 17 mM $Na_2EDTA$, and 41 mM citric acid. A transfection reagent (Metafectene Pro) was purchased from Biontex (Plannegg, Germany).

## Bioinformatics and sequence analysis

Toll sequences of *Bombyx mori* [50] were used as queries to obtain *S. exigua* Tolls (*Se-Tolls*) from the transcriptome of *S. exigua* with the following GenBank accession numbers: GGRZ01034653.1 for *Se-Toll1*, GGRZ01152585.1 for *Se-Toll2*, GGRZ01091454.1 for *Se-Toll3*, GGRZ01062293.1 for *Se-Toll4*, GGRZ01239670.1 for *Se-Toll5*, GGRZ01119041.1 for *Se-Toll6*, GARL01056056.1 for *Se-Toll7*, GGRZ01148168.1 for *Se-Toll8*, GGRZ01158569.1 for *Se-Toll9*, and GGRZ01242837.1 for *Se-Toll10*. Using the same method, *S. exigua* Spätzles (*Se-Spz*) genes were obtained from GenBank (accession numbers of GGRZ01098176.1 and GGRZ01162530.1 for *Se-Spz1* and *Se-Spz2*, respectively). Phylogenetic relationship and domain prediction analyses were performed using MEGA6 and Clustal W programs from EMBL-EBI (www.ebi.ac.uk). Bootstrapping values were obtained with 1,000 repetitions to support branches. Protein domains were predicted using SMART (http://smart.embl-heidelberg.de/) and Pfam (http://pfam.xfam.org).

## RNA extraction and RT-PCR or RT-qPCR

Total RNA extraction, cDNA preparation, and RT-qPCR followed the procedure described by Mollah et al. [48]. Briefly, the synthesized single-stranded cDNA was used as template for RT-PCR with 35 rounds of a temperature cycle of 94˚C for 30 sec, different annealing temperatures for 30 sec, and 72˚C for 30 sec after an initial heat treatment at 94˚C for 2 min with gene-specific primers (S1 Table). RT-qPCR was conducted using a Step One Real-Time PCR System (Applied Biosystem, Marsiling, Singapore) wit the following conditions: 95˚C for 10 min for initial heat followed by 40 cycles of 95˚C for 15 sec, different annealing temperature for 30 Sec, and 72˚C for 30 sec, and 1 cycle of 95˚C for 15 sec, 60˚C for 1 min, 95˚C for 15 sec for dissociation using gene-specific primers (S1 Table). A ribosomal gene, *RL32*, was used as reference gene. Quantitative analysis was done with a comparative CT method as reported by Livak and Schmittgen [51] to estimate mRNA expression levels. Each experiment was replicated three times.

## RNA interference (RNAi) for Se-Toll and Se-Spz expression

Template cDNA was amplified using *Se-Toll* or *Se-Spz* primers (S1 Table) containing T7 RNA polymerase promoter sequence (5´-TAATACGACTCACTATAGGGAGA-3´) at 5´ ends.

These amplified PCR products were used for double-stranded RNA (dsRNA) synthesis using a MEGAscript RNAi kit (Ambion, Austin, TX, USA) based on the manufacturer's instructions. Control dsRNA was synthesized from a viral gene, *CpBV302* [52]. Purified dsRNA was injected to the hemocoel of *S. exigua* larvae (1 μg/larva) after mixing with the same volume of transfection reagent and incubating at 25°C for 30 min for liposome formation. From this mixture, 1 μg of dsRNA in 2 μL volume was micro-injected into *S. exigua* larval hemocoel. RNAi efficacy for reducing *Se-Toll* or *Se-Spz* expression was determined by RT-qPCR at 0, 12, 24, 48, and 72 h post-injection (PI). At 24 h PI, treated larvae were used for immune challenge experiments.

## Extraction of bacterial secondary metabolites and thin layer chromatography (TLC)

*X. hominickii* was cultured in 1 L of TSB at 28°C for 48 h. After centrifuging cultured broth at $10,000 \times g$ for 20 min at 4°C, the resulting supernatant was used for organic extraction as described by Mollah et al. [48]. Briefly, the supernatant was mixed with the same volume of hexane. After 30 min of incubation 4°C, the hexane extract (HEX) was separated from the aqueous fraction. The same procedure was sequentially used to obtain chloroform (CX), ethylacetate (EAX), and butanol (BX) extracts. Resulting organic extracts containing bacterial metabolites were dried with a rotary evaporator (Eyela N-1110, Rikakikai, Tokyo, Japan). After weighing, extracts were resuspended in methanol. TLC was performed for resulting extracts to obtain metabolites on a silica gel plate (20×20 cm; Merck, Darmstadt, Germany). Different compositions of chloroform and methanol (v/v) were used as eluents. The developed silica gel plate was incubated with a mixture (19:1, g/g) of sea sand (Merck) and iodine (Duksan, Ansan, Korea). Spots were visualized and marked in a fluorescence analysis cabinet (Spectroline, CM-10, Westbury, NY, USA).

## Fractionation of bacterial secondary metabolites using column chromatography

Potent organic extract in the binding assay (see below) was subjected to column chromatography filled with silica gel 60 (0.063–0.200 mm; Merck) using a chloroform/methanol eluent with increasing amount of methanol from 100:0 to 0:100 (v/v) (S2 Fig). Each resulting fraction was dried and dissolved in methanol for binding assay with a recombinant Se-DSP1. Active fractions were further separated by a preparatory thin layer chromatography (Merck) with chloroform: methanol: acetic acid (7.5:2:0.5, v/v). Resulting potent subfractions were subjected to GC-MS analysis for compound identification.

## Chemical determination using gas chromatography-mass spectrometry (GC-MS)

GC-MS analysis of potent bacterial fractions followed the method described by Mollah et al. [7]. Briefly, an MS (5977A Network, Agilent Technologies; Santa Clara, CA, USA) was coupled with GC (7890B, Agilent Technologies) equipped with a non-polar column (HP5 MS column, Agilent Technologies). The carrier gas was helium at a flow rate of 1 mL/min. The injector temperature was set at 200°C. Each potent active fraction from binding assays was dissolved in methanol. The result suspension (1 μL) was injected at a split mode with a split ratio of 10:1. The oven temperature was initiated at 100°C for 3 min and then raised to 300°C at a rate of 5°C/min. The final temperature (300°C) was continued for 10 min. The total run-time was 53 min. Mass spectra were recorded in EI mode at 70 eV with a scanning range of 33–550 m/z.

Samples were identified by comparing their mass spectra with those in the database (NIST 11, Version 2.0, NIST, Gaithersburg, MD, USA).

## Preparation of recombinant Se-DSP1 (rSe-DSP1) protein

Recombinant *Escherichia coli* [11] expressing *Se-DSP1* was grown at 37˚C for 4 h with shaking at 200 rpm. Soluble recombinant protein was extracted from *E. coli* pellet using a sonication after two freezing-thawing cycles with liquid nitrogen. After centrifugation at $14,000 \times g$ for 15 min at 4˚C, the resulting supernatant was used for protein purification. A native condition method was used to purify 6xHis-tagged soluble proteins. The supernatant (10 mL) obtained from the previous step was mixed with 1 mL of Ni-NTA agarose resin (Qiagen, Hilden, Germany) in 15 mL Econo-Pac Chromatopgaphy column (1.5 ×12 cm, BioRad, Hercules, CA, USA) and agitated for 60 min at 4˚C on a rocker platform. The resin containing rSe-DSP1 was washed four times with 10 mL of native wash buffer (50 mM potassium phosphate, pH 8.0, 500 mM NaCl, and 20 mM imidazole). Finally, proteins remained in the resin of the column were eluted with 10 mL of native elution buffer (50 mM $NaH_2PO_4$, 500 mM NaCl, 250 mM imidazole, pH 8.0). Presence or absence of purified protein was detected using 10% SDS-PAGE and confirmed by western blotting. Dialysis was done using a semipermeable tube (3 kDa cut-off size, Sigma-Aldrich Korea) against PBS by changing fresh buffer three times at 6 h intervals. Finally, the protein solution was concentrated using a TFD5503 Bench-Top freeze dryer (Ilshine, Seoul, Korea) overnight and dissolved in 1 mL of PBS. Protein solution was then kept at -20˚C.

## Binding of *X. hominickii* metabolites with rSe-DSP1 using thermal shift assay

rSe-DSP1 binding to *X. hominickii* metabolites was evaluated using a thermal shift assay [53] with a Protein Thermal Shift dye kit (Applied Biosystem, Foster City, CA, USA) according to the manufacturer's instruction. Binding assay was conducted as described by Mollah et al. [11]. Briefly, a reaction mixture consisted of 5 μL of protein thermal shift buffer, 2.5 μL of protein thermal shift dye, 10 μL of rSe-DSP1 (500 ng), and 2.5 μL of test metabolite at different final concentrations (0, 2, 4, 6 and 8 μM). A melting curve experiment type was set up using a Step One real time PCR system (Applied Biosystems, Foster City, CA, USA). Thermal profile was obtained at 25˚C for 2 min and 99˚C for 2 min. Melting temperatures resulting from the experiment were plotted with SigmaPlot 10.0 (Systat Software, San Jose, CA, USA). Dissociation constant (Kd) was calculated using ligand binding equation category.

## Immunofluorescence assay

For immune challenge, $4 \times 10^5$ cells of *E. mundtii* were injected into L5 larva. For inhibitor assay, 1 μg of EMP was injected into L5 larva. At 6 h PI, the hemolymph ($\sim$250 μL) was collected into 750 μL of ACB and incubated on ice for 30 min. After centrifugation at $800 \times g$ for 2 min, 800 μL of supernatant was discarded and 250 μL of TC100 insect tissue culture medium (Welgene, Gyeongsan, Korea) was added. Fat body was collected from the insect hemocoel into ACB after removing whole gut. The same procedure was used to isolate hemocytes. Tissue suspension (10 μL) was loaded onto a glass coverslip and incubated in a wet chamber for 30 min in a dark condition. Cells were then fixed with 4% formaldehyde for 10 min at room temperature (RT). After washing thrice with PBS, cells were permeabilized with 0.2% Triton X-100 in PBS for 2 min at RT. Cells were washed thrice in PBS and blocked with 3% BSA in PBS for 10 min. After washing once with PBS, cells were incubated with Alexa Fluor 555 phalloidin and a primary antibody (Abclone, Seoul, Korea) raised against Se-DSP1 for 1 h 20 min at RT.

After washing thrice, cells were incubated with a secondary antibody conjugated with FITC (Sigma-Aldrich Korea) for 1 h. Following three washings with PBS, cells were incubated with 4′,6-diamidino-2-phenylindole (DAPI, 1 μg/mL) (Thermo Scientific, Rockford, IL, USA) in PBS for nucleus staining. Finally, after washing thrice with PBS, cells were adhered to slide glass and observed under a fluorescence microscope (DM2500, Leica, Wetzlar, Germany) at 400 × magnification.

## Exosome isolation and western blotting

Exosome was extracted using an ExoQuick (System Biosciences, Palo Alto, CA, USA) according to the manufacturer's instruction. Briefly, 500 μL of hemolymph was collected from 20 L5 larvae and centrifuged at 3,000 × g for 15 min. The supernatant (250 μL) was mixed with 63 μL of ExoQuick solution and incubated at 4˚C for 30 min. Then the mixture was centrifuged at 1,500 × g for 30 min at 4˚C. A white exosome pellet was resuspended in PBS. For western blot analysis, extracted exosome proteins (20 μg per sample) were separated on 10% SDS-PAGE. These separated samples in the gel were transferred onto 0.2 μm pore nitrocellulose membranes (BioRad) for 45 min at 100 V in chilled transfer buffer (25 mM Tris base, 190 mM glycine, 20% methanol, pH 8.5). Membranes were briefly rinsed with Tris-buffered saline containing Tween-20 (TBST) (20 mM Tris, 150 mM NaCl, and 0.1% Tween 20, pH 7.5) and then blocked with 3% bovine serum albumin (BSA) in TBST at RT for 1 h. Membranes were then incubated with an anti-CD9 Rabbit monoclonal antibody (Cell Signaling Technology, Danvers, MA, USA) specific to exosome as a primary antibody diluted 1,000 times with TBST containing 3% BSA at 4˚C for 2 h. Membranes were then washed three times with TBST (5 min per washing) and then incubated with an anti-Rrbbit IgG-alkaline phosphatase secondary antibody (Sigma-Aldrich Korea) at a dilution of 1:2,000 in TBST containing 3% BSA for 1 h at RT. Blots were rinsed three times with TBST. To detect alkaline phosphatase activity, nitrocellulose membrane blots were incubated with a substrate (BICP/NBT, Sigma-Aldrich Korea).

## PLA$_2$ and PO enzyme activity assay

For measuring PLA$_2$ enzyme activity of the treated larvae sample, PLA$_2$ Assay Kit (Cayman Chemical, Ann Arbor, MI, USA) was used as described by Vatanparast et al. [54]. Activity of phenoloxidase (PO) from plasma was determined using the method described by Sadekuzzaman et al. [23]. Briefly, L-3,4-dihydroxyphenylalanine (DOPA) was used as substrate to check the enzyme activity of PO in the plasma collected from treated larvae. A reaction volume of 200 μL consisted of 180 μL of 10 mM DOPA in PBS and 20 μL of the plasma. Absorbance was taken at 490 nm using a VICTOR multi label Plate reader (PerkinElmer, Waltham, MA, USA). PO activity was expressed as ΔABS/min/μL of plasma. Each treatment was replicated three times (each time with three independent samples).

## Construction of a deletion mutant using CRISPR/Cas9

The structure of *Se-Toll9* genomic DNA was analyzed after sequencing the cognate gDNA. After confirming no intron, the exon sequence was submitted to an online tool (http://www.chopchop.com) to determine the optimal target site. Based on a target containing a protospacer adjacent motif (PAM), two 20 bp (5′-AGATCGGAGTTTCCGTATCGAGG-3′ and 5′-ATAGATTAAGTATAACCATACGG-3′) targeting sites were selected for sgRNA1 and sgRNA2, respectively. sgRNA was generated using Guide-it sgRNA In Vitro Transcription kit (Takara Korea Biomedical, Seoul, Korea) according to the manufacturer's manual. Briefly, PCR was performed using target oriented forward primer and company-provided reverse

primer. PCR conditions included a pretreatment at 98˚C for 5 s and subsequent amplification with 33 cycles of 98˚C for 10 s and 68˚C for 10 s. The resulting PCR product was used for *in vitro* transcription using T7 RNA polymerase to produce sgRNA, which was purified using a spin column provided by the kit. The final amount was quantified with a spectrophotometer (NanoDrop, Thermo Fisher Scientific Korea, Seoul, Korea). Females were kept in the dark to lay eggs on a kitchen paper towel for 1 h. Eggs were dried in a desiccator for 10 min at RT. Dried eggs were then fixed on a cover slip with a double-sided tape. Before injection, glass capillaries (10 μL quartz, World Precision Instrument, Sarasota, FL, USA) with sharp points (< 20 μm diameter) were prepared with a Narishige magnetic glass microelectrode horizontal puller model PN30 (Tritech Research, Los Angeles, CA, USA). Eggs were injected through a micropyle using a Sutter $CO_2$ Pico pump injector (PV830, World Precision Instrument) under a stereomicroscope (SZX-ILLK200, Olympus). The injection volume per egg was 10 nL of a mixture containing Cas9 (500 ng/μL) and sgRNA (50 ng/μL). All injections were finished within 30 min after egg collection, including drying time. Injection-treated eggs were incubated at RT for 4 h before transferring to a growing chamber (25˚C). They were then observed for 4 days until hatching. Genomic DNA extraction, PCR, and sequencing were carried out to detect mutant insects using gene-specific primers (S1 Table) producing a 808 bp product. Genomic DNA was extracted using 10% Chelex (Biorad, Hercules, CA, USA) from hemolymph of L5 larvae (~10 μL/larva). PCR conditions had a pretreatment at 94˚C for 2 min, an amplification step with 35 cycles of 94˚C for 1 min, 56.8˚C for 1 min, and 72˚C for 1 min, and a final extension step at 72˚C for 10 min. PCR products were then cloned into pCR2.1 vector (Thermo Fisher Scientific Korea) and bidirectionally sequenced.

## Bioassay of RNAi-treated *S. exigua* against *Bacillus thuringiensis* (Bt)

To assess Bt virulence to RNAi-treated *S. exigua*, L5 larvae were injected with 1 μg of dsRNA. At 24 h after dsRNA injection, feeding assay was applied with a leaf-dipping method. Briefly, a piece of cabbage leaf (3 × 3 cm) was soaked in 500 ppm of Bt suspension for 5 min. Treated leaves were then provided to dsRNA treated larvae for 24 h. Treated larvae were then incubated for another 2 days under rearing conditions to observe mortality. Each replication consisted of 10 larvae. Each treatment was replicated three times. A commercial product of *B. thuringiensis* var. *kurstaki* (serotype IIIa & IIIb, Hanearl Science Corporation, Taebaek, Korea) was used for mortality test at concentration of 500 ppm.

## Statistical analysis

All data for continuous variables were subjected to one-way analysis of variance (ANOVA) using PROC GLM in SAS program [55]. Mortality data were subjected to arcsine transformation and used for ANOVA. Means were compared with the least significant difference (LSD) test at Type I error = 0.05. Median lethal dose ($LD_{50}$) was subjected to Probit analysis using EPA Probit Analysis Program, ver. 1.5 (Environmental Protection Agency, USA).

## Supporting information

**S1 Table. List of primers used in this study.**
(DOCX)

**S2 Table. GenBank accession numbers used for phylogenetic analysis.**
(DOCX)

**S1 Fig. Expression profile of 10 Toll genes ('T1-T10') of *S. exigua*.** (A) Egg developmental stage. (B) Larval hemocytes. (C) Larval midgut. A ribosomal RNA, *RL32*, was used as

endogenous control. Expression level was calculated as fold change from the lowest expression value. Each treatment was independently replicated three times.
(DOCX)

**S2 Fig. RNAi efficiencies of 10 Se-Toll genes by injecting gene-specific dsRNAs ('dsToll1-dsToll10', 1 μg/larva) to L5 larvae.** A viral gene, *CpBV302*, was used to prepare control dsRNA ('dsCON'). Expression level of a ribosomal gene, RL32, was used as reference to normalize expression levels of target genes. Each treatment was independently replicated three times.
(DOCX)

**S3 Fig. RNAi efficiencies of two Se-Spz genes by injecting gene-specific dsRNAs ('dsSpz1 and dsSpz2', 1 μg/larva) to L5 larvae.** A viral gene, *CpBV302*, was used to prepare control dsRNA ('dsCON'). Expression level of a ribosomal gene, *RL32*, was used as reference to normalize expression levels of target genes. Each treatment was independently replicated three times.
(DOCX)

**S4 Fig. A diagram illustrating fractionation steps of culture broth of *X. hominickii*.** Organic extracts were obtained using hexane ('HEX'), ethyl acetate ('EAX'), chloroform ('CX'), and butanol ('BX'). BX was fractionated using a chromatography column filled with silica gel where a gradient chloroform/methanol mixture with increasing amount of methanol from 100:0 to 0:100 (v/v) was used. Active butanol fractions were separated using a preparatory thin layer chromatography ('TLC').
(DOCX)

**S5 Fig. GC-MS analysis of active fractions and prediction of compounds from *X. hominickii* culture broth extracted by butanol.** EMP, 3-ethoxy-4-methoxy phenol; HMPP, hexahydro-3-(2-methylpropyl)-pyrrolo[1,2-a]pyrazine-1,4-dione; BEP, bis (2-ethylhexyl) phthalate; IND, indole; DBA, dibutylamine; PM, phthalimide; CBA, o-cyanobenzoic acid.
(DOCX)

## Acknowledgments

We thank Miltan Roy for assistance with bacterial culture and Dooyeol Choi for his assistance in statistical analysis. We also appreciate Youngim Song for her supplying experimental materials.

## Author Contributions

**Conceptualization:** Yonggyun Kim.

**Data curation:** Md. Mahi Imam Mollah, Shabbir Ahmed.

**Formal analysis:** Md. Mahi Imam Mollah, Shabbir Ahmed.

**Methodology:** Md. Mahi Imam Mollah, Shabbir Ahmed, Yonggyun Kim.

**Resources:** Yonggyun Kim.

**Validation:** Md. Mahi Imam Mollah, Shabbir Ahmed, Yonggyun Kim.

**Writing – original draft:** Md. Mahi Imam Mollah, Shabbir Ahmed, Yonggyun Kim.

**Writing – review & editing:** Yonggyun Kim.

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
