## [Decision Letter · Decision Letter 0]

14 Feb 2021

Dear Dr. Kim,

Thank you very much for submitting your manuscript "Immune mediation of HMG-like DSP1 via Toll-Spätzle pathway and its specific inhibition by salicylic acid analogs" for consideration at PLOS Pathogens. As with all papers reviewed by the journal, your manuscript was reviewed by members of the editorial board and by several independent reviewers. The reviewers appreciated the attention to an important topic. Based on the reviews, we are likely to accept this manuscript for publication, providing that you modify the manuscript according to the review recommendations.

Sincerely,

Francis Michael Jiggins

Associate Editor

PLOS Pathogens

Karla Satchell

Section Editor

PLOS Pathogens

Kasturi Haldar

Editor-in-Chief

PLOS Pathogens

orcid.org/0000-0001-5065-158X

Michael Malim

Editor-in-Chief

PLOS Pathogens

orcid.org/0000-0002-7699-2064

Reviewer Comments (if any, and for reference):

Reviewer's Responses to Questions

**Part I - Summary**

Reviewer #1: The manuscript characterizes the mechanisms by which X. hominikii suppresses the immune response in the insect S. exigua. It identified the genes involved and uses both ectopic expression and dsRNA knockdown and/or CRISPR mutants to define the roles of DSP1, Toll receptors, and Spatzle (a Toll ligand) in the immune response in S. exigua. It goes on to define the secondary metabolite from X. hominikii that suppresses this immune response. These findings provide significant insight into how x. hominikii causes immunosuppression. The manuscript is well-written, and the data presented are generally clear and convincing. However, there are a few areas that could be modified to improve the clarity.

Reviewer #2: This study investigated the immune signaling pathway of a damage-associated molecular pattern, DSP1. It reports that DSP1 activates Toll/Spz pathway for defending Gram-positive bacterial infection. Especially, Toll9 among 10 Toll receptors was a main signal component. This is further supported by a deletion mutant of Toll9, which lost immune activation. Furthermore this study identified an immune blocker, EMP, which exhibited a tight binding to DSP1 not to be released. Thus the entomopathogenic bacterium, X. hominickii induces a fatal pathogenicity by releasing EMP to suppress host immune defense. Overall, this reports a new immune signaling pathway in insects using a DAMP signal, DSP1 and a novel immunosuppressive agent, EMP. Thus, this study fits to the journal scope and standard of PLoS Pathogens. However, following issues need to be addressed.

1. This study showed the release of DSP1 from nucleus to plasma. The released DSP1 appears to be transported in plasma using exosome cargo. Here are questions. What does the intracellular signal stimulate DSP1 release from nucleus? Why does the DSP1 take exosome in the plasma?

2. In addition to Toll9, DSP1 may use Toll6, Toll7, or Toll8 to induce the AMP or PO activation from Fig. 4. Why did you determine Toll9 as DSP1 signal component?

3. I am not sure that DSP1 directly activated Spz to trigger Toll sigaling pathway?

4. What is the relationship of Bt pathogenicity and immunosuppression? Is it required for the Bt to exhibit its insecticidal activity?

5. How can you explain the inhibitory activity of EMP on SP1 release from the nucleus?

Reviewer #3: Precis

Some insect pathogenic microbes overcome host immune responses to infection by secreting compounds that inhibit insect immune reactions to infection. Park and Kim (2000; doi: 10.1016/s0022-1910(00)00071-8) first reported that the bacterial lethality of the insect pathogen Xenorhabdus nematophilus was attenuated after injecting arachidonic acid (AA), a direct precursor to biosynthesis of prostaglandins and other eicosanoids, into the abdomens of infected lepidopterans, Spodoptera exigua. They put forth the idea the bacterium suppresses insect immunity by inhibiting biosynthesis of eicosanoids. Now, after 20 years and many publications on eicosanoid signaling in insect immunity, they turned attention to a related insect pathogen, X. hominickii. This pathogen also secretes metabolites that inhibit the first step in eicosanoid biosynthesis, phospholipase A2 (PLA2), which effectively suppresses eicosanoid signaling to suppress host immunity. They reported that a damage-associated molecular pattern called dorsal switch protein 1 (DSP1), which activates PLA2 early in infection in S. exigua. They used immunofluorescence to record increased hemocyte spreading and Se-DSP1 in fat body. They used western blots to show Se-DSP2 in hemolymph of infected, but not naïve larvae at 6 h post-infection. Their western blot analysis also indicated the Se-DSP1 was transported out the cells as via exosomes. The authors created a recombinant Se-DSP1 (rSe-DSP1) and used it to demonstrate that bacterial infection with another insect pathogen, E. mundtii, leads to increased PO activity, as does rDSP1 treatments, but not after denaturing the protein. Treating larvae with a dsRNA construct, dsDSP1 + E. mundtii blocked infection-triggered PO activity. They used enzyme activity assays to show rDSP1 treatments led to significant increases in secretory PLA2 and cellular PLA2 activities. Translating to the influence of rDSP1 on humoral immunity, the authors show that rDSP1 injections led to significant increases in mRNAs encoding a range of antimicrobial peptides and proteins, including apolipophoren-III, cecropin and lysozyme. They report ten Toll receptors and show accumulations of mRNAs encoding them. Expression of each of the ten genes was substantially reduced at 24 or 48 h after injecting dsRNA constructs specific to each gene. Co-injections with each dsRNA + rDSP1 led to substantial PO activity, sPLA2 activity and cPLA2 activity. Similar co-injections led to substantial expression of genes encoding three anti-microbial proteins. Digging deeper into humoral immunity, the authors report two genes encoding Spätzels, showing gene structures and a phylogenetic tree placing them withing Diptera-Lepidoptera. They show the genes are expressed in hemocytes and fat body. rDSP1 treatments led to increased gene expression in hemocytes, but not in fat body. Bacterial injections led to increased expression of both genes in fat body. They show dsRNA treatments led to decreased expression of both genes encoding Spätzes from 24 – 72 h post-injections. Bacterial injection and DSP1 injections led to increased PO activity, but not to increased PLA2 activity. Co-inections with dsSpätze1 + DSP1 did not lead to increased expression of three anti-microbial peptides, although dsSpätze2 +DSP1 did.

The authors continued drilling into humoral immune signaling by creating CRISPR/Cas9 deletion mutants in SeToll9. Although rDSP1 injections into wild-type larvae led to increased sPLA2 and cPLA2 activities and to increased expression of the three anti-microbial proteins, similar injections into the deletion mutants did not. The changes in immune parameters just mentioned translated into increased mortality following co-injections with most dsRNAs designed to each of the ten toll receptors + the insect pathogen, Bacillus turingiensis Mortality did not increases following dsTOLL1, 8 or 10. Similarly, dsSpätze1 and -2 treatments also led to increased mortality. The authors identified X. hominickii metabolites that bind to rSeDSP1, and thus, block its translocation. They used GC-MS analysis to identify several metabolites with binding affinities in the low �M range, of which 3-eethoxy-4-methoxyphenol (EMP) had the highest binding affinity. In a direct test of DSP1 translocation, the authors presented a western blot showing the presence of Se-DSP1 in plasma from larvae injected with the pathogen E. mundtii, but not in larvae injected with the pathogen + EMP. EMP + rDSP1 treatments led to sharp reductions in PO, sPLA2 and cPLA2 activities and virtually eliminated accumulations of mRNAs encoding gallerimycin, gloverin and lyzozyme. Injections of pathogen B. thuringiensis led to increased mortality in control larvae. The bacterial injections into larvae treated with dsDSP1 led to higher mortality and still higher mortality followed in larvae treated with EMP.

The authors complete their story with a meaningful model of a hypothesized mechanism of DSP1 action. In their view, the bacterial product EMP damages hemocytes and DSP1 is translocated from cells into hemolymph, where it activates one or more serine protease cascades that leads to active phenol oxidase and melanization reactions. It also activates two Spätzels, with interacts with toll receptors that lead to biosynthesis of eicosanoids that mediate cellular immune reactions and lead to expression of genes encoding lysozyme and other anti-microbial protains.

They present three supplementary figures. Fig. S1 reports accumulations a mRNAs encoding the 10 toll receptors. Fig. S2 shows their extraction scheme for analysis of the X. hominickii metabolites and Fig. S3 shows the chemical structures of selected metabolites.

Critique

The authors present a well-reasoned and thorough series of experiments to test their hypothesis.

**Part II – Major Issues: Key Experiments Required for Acceptance**

Reviewer #1: n/a

Reviewer #2: (No Response)

Reviewer #3: (No Response)

**Part III – Minor Issues: Editorial and Data Presentation Modifications**

Reviewer #1: 1. Abstract: The abstract is incredibly long and it seems like much of this information belongs in the Introduction. Further, from the abstract, the reader expected eicosanoid biosynthesis to be studied in the manuscript and it was not.

2. Line 104 – explain to the reader what Pelle kinase is.

3. Paragraph starting at line 107 – it is not clear what the authors think the order of events is.

4. Figure 1 – The change in Se-DSP1 immunofluorescence isn’t convincing. It seems like it is relocalizes, perhaps to the nucleolus, but the intensity does not appear reduced. Some quantification is needed to clarify this.

5. The terminology in the results and in the figures is not always consistent. For example, Line 134-135, Se-DSP1 (rSe-DSP1) is not how it is written in the Figure 2. Another example is in relation to Figure 7

6. The authors need use PLA2 activity to indicate eicosanoid biosynthesis, but PLA2 has many other roles. The authors need to discuss this limitation. Further, in the results it is not discussed what samples are used for the analysis and why they are the correct samples.

7. It seems it is worth discussing why AMP genes are activated by heat inactivated rSe-DSP1

8. The description of the data in Figure 4 is difficult to follow. Sometimes the authors say which Toll receptors have no effects and for other data which Toll receptors play a role. It would be clearer to the reader to talk about the Toll receptors that are involved in the process. Also a discussion of the difference in the Toll receptor role in PLA activity vs AMP gene expression is warranted. It is unclear what the reader should conclude about who is involved.

9. Line 183 – the data in figure 5c would say that neither Spz gene is induced in the hemocytes in response to EM, and Spz1 may be inhibited. This is not what is stated in the results and it seems worth talking about. Lines 184-185 seems to contradict the prior sentence

10. In the section starting at 209, it would help the reader for the authors to explain their method of assessing pathogenicity.

11. Figure 1 legend talks about Tubulin western blots that are not shown in the figure

12. The RNAi knockdown data in Figure 4a and Figure6a are shown as connected lines. Such a graph implies the expression was assessed repeatedly in the same cells, when in reality it was done on different cells. The data should be presented as individual data points or bar graphs.

Reviewer #2: (No Response)

Reviewer #3: (No Response)

PLOS authors have the option to publish the peer review history of their article (what does this mean?). If published, this will include your full peer review and any attached files.

Reviewer #1: No

Reviewer #2: No

Reviewer #3: **Yes: **David Stanley
---

## [Editor Report · Decision Letter 1]

9 Mar 2021

Dear Dr. Kim,

Thank you very much for submitting your manuscript "Immune mediation of HMG-like DSP1 via Toll-Spätzle pathway and its specific inhibition by salicylic acid analogs" for consideration at PLOS Pathogens. Thank you for thoroughly revising the manuscript, it is a great paper. I am happy with all the revisions in the main text, but would like you to look again at the first comment about the abstract. At the moment the abstract is hard to read unless you have technical knowledge of this area. This is because it includes a large amount of technical detail and goes through many different experiments. In contrast, it does not make clear the wider significance and importance of the work. This contrasts with the author summary, which did this job very well. I would ask that the abstract is simplified and makes clearer the wider significance of your work. I apologise for returning the manuscript a second time, but I hope this will be a quick change to make that will help your work reach a wider audience.

Sincerely,

Francis Michael Jiggins

Associate Editor

PLOS Pathogens

Karla Satchell

Section Editor

PLOS Pathogens

Kasturi Haldar

Editor-in-Chief

PLOS Pathogens

orcid.org/0000-0001-5065-158X

Michael Malim

Editor-in-Chief

PLOS Pathogens

orcid.org/0000-0002-7699-2064

Thank you for thoroughly revising the manuscript which is a very interesting body of work. I am happy with all the revisions in the main text, but would like you to look again at the first comment about the abstract. At the moment the abstract is not suitable for PlOS Pathogens as it is very hard to read unless you have a technical knowledge of this area. This is because it includes a large amount of technical detail and goes through many different experiments. However, it does not make clear the wider significance and importance of the work. This contrasts with the author summary, which did this job very well. I would ask that the abstract is simplified and makes clearer the wider significance of your work.

Reviewer Comments (if any, and for reference):

Figure Files:

Data Requirements:

Reproducibility:

References:

---

## [Editor Report · Decision Letter 2]

11 Mar 2021

Dear Dr. Kim,

We are pleased to inform you that your manuscript 'Immune mediation of HMG-like DSP1 via Toll-Spätzle pathway and its specific inhibition by salicylic acid analogs' has been provisionally accepted for publication in PLOS Pathogens.

Best regards,

Francis Michael Jiggins

Associate Editor

PLOS Pathogens

Karla Satchell

Section Editor

PLOS Pathogens

Kasturi Haldar

Editor-in-Chief

PLOS Pathogens

orcid.org/0000-0001-5065-158X

Michael Malim

Editor-in-Chief

PLOS Pathogens

orcid.org/0000-0002-7699-2064
---

## [Editor Report · Acceptance letter]

18 Mar 2021

Dear Dr. Kim,

We are delighted to inform you that your manuscript, "Immune mediation of HMG-like DSP1 via Toll-Spätzle pathway and its specific inhibition by salicylic acid analogs," has been formally accepted for publication in PLOS Pathogens.

Best regards,

Kasturi Haldar

Editor-in-Chief

PLOS Pathogens

orcid.org/0000-0001-5065-158X

Michael Malim

Editor-in-Chief

PLOS Pathogens

orcid.org/0000-0002-7699-2064